# SPATIO-TEMPORAL DEPENDENCY-AWARE NEURON OPTIMIZATION FOR SPIKING NEURAL NETWORKS

## ABSTRACT

As a biologically inspired computing paradigm, Spiking Neural Networks (SNNs) process information through discrete spike sequences, mimicking the brain's temporal dynamics and energy efficiency. The combination of backpropagation through time (BPTT) and direct input encoding (i.e., feeding decimal data directly into the network) has emerged as the mainstream training approach for SNNs. However, this combination introduces varying temporal dependency requirements across the network's spatial dimension. These differences are often neglected in existing studies, which typically apply uniform temporal dependency configurations throughout the network. Consequently, this could result in missing key gradients or introducing redundant ones in the temporal dimension, ultimately affecting the network's performance. To address this gap, we propose a novel Spatio-Temporal Dependency-Aware Neuron Optimization (ST-DANO) method for SNNs, which consists of two key components: neuron design and neuron search. Specifically, to overcome the limitations of traditional Leaky Integrate-and-Fire (LIF) neurons in adapting to varying temporal dependencies, we designed two variants, Long-LIF and Short-LIF, which improve the neuron's ability to capture long-term and short-term dependencies, respectively, by dynamic modulation of membrane potential thresholds and time constants. After validating our neuron designs through ablation studies, we developed a layer-wise neuron search strategy that automatically selects the optimal neuron type for each layer to ensure optimal temporal dependency configurations across the network. Extensive experiments on static and neuromorphic datasets demonstrate that ST-DANO can effectively adapt to temporal dependency differences across the spatial dimension in SNNs under various time-step configurations. The resulting architectures surpass state-of-the-art performance, achieving a remarkable 83.90% accuracy on the DVS-CIFAR-10 dataset—a more than 5% improvement over the baseline.

## 1 INTRODUCTION

Spiking Neural Networks (SNNs) diverge from traditional paradigms by processing information through discrete spikes, enabling them to capture the temporal dynamics while significantly reducing energy consumption (Ding et al., 2022). While conventional Artificial Neural Networks (ANNs) merely process static data in a single flow, SNNs handle information through binary spike sequences and integrate temporal features across multiple time steps. This distinctive processing method endows SNNs with the ability to handle complex temporal patterns; however, it complicates the use of backpropagation, rendering conventional training methods inapplicable (Neftci et al., 2019).

To address this challenge, Backpropagation Through Time (BPTT) with surrogate gradients was introduced (Wu et al., 2018). BPTT is a spatio-temporal propagation method that unfolds the network across both spatial and temporal dimensions, with gradients propagating from the output layer to the input layer but also recursively accumulating along the time axis (Meng et al., 2023). Building on BPTT, Rueckauer et al., 2017 proposes direct input encoding to feed decimal data directly into the network. Compared to traditional Poisson (Heeger et al., 2000) and temporal encodings (Mostafa, 2017), direct input encoding significantly reduces the number of time steps required for information representation, lowering both training and inference costs. This efficiency gain has made the combination of BPTT and direct input encoding one of the mainstream training methods in recent research (Deng et al., 2022; Meng et al., 2023; Xiao et al., 2022; Wang et al., 2023a).

However, this combination introduces varying requirements for temporal dependencies across the network spatial dimension—an aspect overlooked by existing research. This phenomenon stems from two key factors. First, BPTT operates by updating network weights only after all time steps have been processed, keeping the weights constant throughout the entire forward pass (Dampfhoffer et al., 2023). Second, in direct input encoding, the inputs to the network remain constant at each time step. These two constant conditions cause shallow[1] neurons to receive nearly identical input across time steps, resulting in diminished temporal dynamics. As the network deepens, the influence of these constant conditions weakens due to the increasing introduction of continuous-to-discrete transformations and nonlinear operations. As a result, deeper neurons are tasked with processing more abstract temporal features, thereby exhibiting much stronger temporal dynamics. This spatial variation in temporal dynamics indicates that neurons at different layers of the network have varying requirements for temporal dependencies. However, existing research based on BPTT and direct input encoding often employs the same type of neuron throughout the entire network, assuming that temporal dependency needs of neurons at all layers are identical, overlooking these critical varying requirements. This approach may lead to the loss of key gradients or the introduction of redundant gradients across the temporal dimension, ultimately constraining network performance.

To address this limitation, we propose a Spatio-Temporal Dependency-Aware Neuron Optimization (ST-DANO) method for SNNs, which optimally adapts temporal dependencies along the spatial dimension. ST-DANO consists of two key modules, neuron design and neuron search. Specifically, to address the limitations of traditional Leaky Integrate-and-Fire (LIF) neurons in adapting variable temporal dependencies, we designed two novel neurons: Long-LIF and Short-LIF, which leverage dynamic modulation of membrane potential thresholds and time constants to adaptively capture and flexibly model long-term and short-term dependencies, respectively. Through ablation studies, we validate the effectiveness of designed neuron variants and investigate how their configurations at different network layers affect overall performance. Next, we propose a layer-wise neuron search strategy that selects optimal neuron types at different network depths, ensuring that temporal dependencies are effectively adapted along the spatial dimension. Extensive experiments on static and neuromorphic datasets demonstrate that the architectures with optimized neuron configurations outperform state-of-the-art models. Remarkably, ST-DANO achieves a top accuracy of 60.24% on Tiny-ImageNet (Deng et al., 2009), while delivering a significant improvement of more than 5% over the baseline on DVS-CIFAR-10 (Amir et al., 2017).

## 2 RELATED WORK

**Training method of SNNs.** There are two mainstream training methods for SNNs: ANN-to-SNN (A2S) conversion and direct training. A2S involves training the network using traditional backpropagation on an ANN, then converting it to an SNN by applying techniques such as activation function mapping or scaling factors (Fang et al., 2021a; Li et al., 2021a; Bu et al., 2023). However, since the training process mirrors that of an ANN, A2S struggles to effectively capture the temporal dynamics inherent in time-series data, limiting its compatibility with neuromorphic datasets (Deng et al., 2020). Direct training methods can be divided into two subcategories: Spike-timing-dependent plasticity (STDP) and backpropagation-based training. As an unsupervised learning method, STDP does not rely on large amounts of labeled data for learning, but its performance is significantly lower than backpropagation-based supervised learning methods (Dong et al., 2023; Zhang et al., 2018). Among backpropagation methods, BPTT with surrogate gradients (SG) dominates, as numerous studies have demonstrated its ability to effectively train high-performance SNNs on both static and neuromorphic datasets (Deng et al., 2022; Wang et al., 2023b). Based on BPTT, various techniques have been proposed to further enhance network performance, such as efficient spatial learning (Meng et al., 2023), normalization methods (Jiang et al., 2024; Guo et al., 2023b), and extra learnable parameters (Sun et al., 2023; Fang et al., 2021b).

**Temporal dependency of SNNs.** Most existing studies on the temporal dependencies of SNNs treat the network as a whole, emphasizing the overall enhancement of long-term dependencies on external inputs. For instance, Lotfi Rezaabad & Vishwanath, 2020 integrates LSTM with SNNs and proposed an error backpropagation method to improve the ability of SNNs to learn long-term dependencies. Stan & Rhodes, 2024 introduces a state space model to SNNs, allowing them to learn

---

[1]In this paper, **shallow** refers to layers closer to the input, while **deep** refers to those closer to the output.

from long sequences. Yu et al., 2022, inspired by biological synapses, develops spatio-temporal synaptic connections to enhance the extraction of spatio-temporal features in SNNs. Gao et al., 2024 proposes spike spatio-temporal attention to strengthen the network's ability to capture temporal information, while Shen et al., 2022 proposes a spatio-temporal adjustment method to enhance the temporal dependencies of SNNs. However, none of these works analyze the inherent variation in temporal dependency demands across different layers after unfolding the network along the spatial dimension. The work most closely related to ours is Kim et al., 2023, which uses Fisher information matrix analysis to show that the concentration of temporal information in SNNs is unevenly distributed across network depths in various tasks. However, this work does not propose a corresponding layer-wise strategy, resulting in limited performance improvement.

## 3  PRELIMINARY

### 3.1  LEAK INTEGRATE-AND-FIRE (LIF) NEURON

This paper employs a widely adopted neuron model, the LIF neuron, which processes temporal information through mechanisms of membrane potential accumulation, spike firing, and reset (Fang et al., 2021b). The dynamics of the LIF neuron can be described as follows.

$$\tau \frac{dV[t]}{dt} = -V[t] + X[t] \tag{1}$$

$$S[t] = \Theta(V[t] - V_{\text{th}}) \tag{2}$$

where $V[t], X[t]$ represent the membrane potential and input current at time step $t$, respectively, and $\tau$ is the membrane time constant. $S[t]$ in Eq. (2) is the output of a LIF neuron, where $V_{\text{th}}$ is the membrane potential threshold and $\Theta$ is the Heaviside function.

### 3.2  GRADIENTS OF BPTT

To investigate the temporal dynamics of a single spiking neuron, this section unfolds the BPTT process and derives the gradient of the neuron's output with respect to its input, providing a theoretical foundation for subsequent design. First, by incorporating the membrane potential hard reset method, Eq. (1) can be recursively written as Eq. (3), where the hard reset process is given in Appendix A.

$$V[t] = V[t-1] \cdot \exp(-\frac{1}{\tau}) \cdot (1 - \Theta(V[t-1] - V_{\text{th}})) + X[t] \tag{3}$$

Based on the above equation, assuming that within $T$ time steps, the input and output sequences of a neuron are $\mathbf{X} = [X[0], X[1], \cdots, X[T]]$ and $\mathbf{S} = [S[0], S[1], \cdots, S[T]]$, respectively. From this, we can derive the gradient of the neuron's output with respect to its input during BPTT as Eq. (4) (the detailed derivation process is provided in Appendix B), where $\kappa_h$ is the derivative of the term $V[x] \cdot (1 - \Theta(V[x] - V_{\text{th}}))$ in the Eq. (3) with respect to $V[x]$.

$$\nabla_{\mathbf{X}}\mathbf{S} = \sum_{t=0}^{T} \sum_{i=0}^{t} \frac{\partial S[t]}{\partial V[i]} \frac{\partial V[i]}{\partial X[i]} = \sum_{t=0}^{T} \Theta'(V[t] - V_{\text{th}}) \cdot (1 + \sum_{i=0}^{t} \exp(-\frac{i}{\tau}) \cdot \prod_{j=1}^{i} \kappa_h(V[t-j])) \tag{4}$$

$$\kappa_h(V[x]) = 1 - \Theta(V[x] - V_{\text{th}}) - V[x] \cdot \Theta'(V[x] - V_{\text{th}}) \tag{5}$$

In the Eqs. (4) and (5), $\Theta'$ represents the derivative of the Heaviside function, and we use the piecewise quadratic function as the surrogate function for it (details are provided in Appendix A).

## 4 METHOD

In this section, we present our ST-DANO. The implementation of ST-DANO has two parts: neuron design and neuron search. Specifically, we design two LIF neuron variants, Long-LIF (LLIF) and Short-LIF (SLIF), to adaptively enhance neurons' long-term and short-term dependencies, respectively. We then propose a layer-wise neuron search strategy based on LIF, LLIF, and SLIF neurons, which selects the most suitable neuron type for each layer in the SNN architecture.

### 4.1 NEURON DESIGN

To enhance long-term and short-term dependencies in LIF neurons, we approached the design from the perspective of gradient contributions. Following the BPTT process, gradients are accumulated after a total of $T$ time steps. For neurons designed to enhance long-term dependencies, the proportion of gradient contributions from early time steps[2] must increase relative to the overall gradient. Conversely, for neurons emphasizing short-term dependencies, the gradient contributions from recent time steps should have a stronger impact. According to the gradient Eq. (4), three key factors influence the gradient: the membrane potential at each time step, the membrane potential threshold $V_{th}$, and the membrane time constant $\tau$. Since the membrane potential is part of the data flow and not suitable for direct modification, we focus on adjusting the adaptive $V_{th}$ and $\tau$ to influence gradient contributions, thereby reinforcing neurons' short-term and long-term dependencies.

Theoretically, a higher $V_{th}$ requires the neuron to accumulate more input before firing, meaning the neuron does not immediately respond to each time step but rather reacts to information accumulated over a longer period. This enables the neuron to integrate context across multiple time steps, thereby improving its capacity to capture long-term dependencies. Conversely, a lower $V_{th}$ increases the neuron's sensitivity to short-term dependencies. As for the membrane time constant $\tau$, a larger value causes the membrane potential to decay more slowly over time, allowing the neuron to retain information from several past time steps, which also aids in capturing long-term dependencies. On the other hand, a smaller $\tau$ enhances the neuron's focus on short-term dependencies by limiting the retention of past information. From the perspective of firing rate, we enhance long-term dependencies by adaptively increasing the membrane potential threshold and strengthen short-term dependencies by reducing the membrane time constant. Both strategies work to suppress spike firing, which not only improves network performance but also reduces energy consumption.

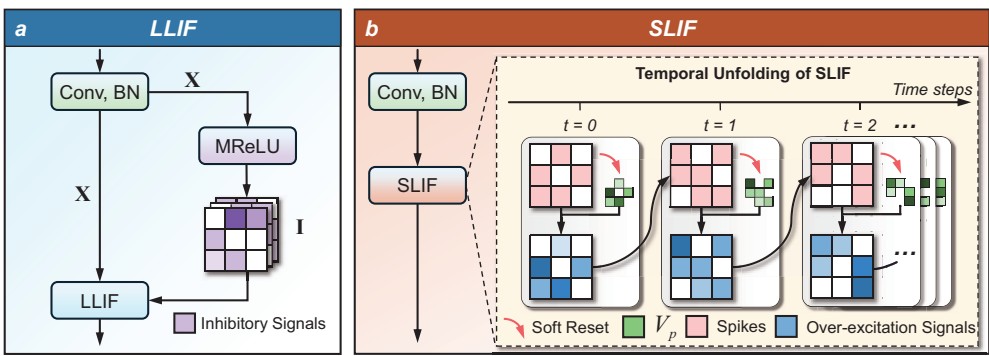

Figure 1: Workflows of (**a**) LLIF and (**b**) SLIF.

**Long-LIF (LLIF).** Since negative signals in the input often represent inhibitory or reversal trends within temporal patterns, in the design of LIF neurons with enhanced long-term dependencies (LLIF), we transform these negative signals at each time step into inhibitory signals, which are then accumulated into the membrane potential threshold, affecting spike firing in all subsequent time steps. This mechanism ensures that the neuron becomes more sensitive to global changes and trends from early time steps, allowing it to accumulate critical information over extended time periods, thereby improving its ability to capture long-term dependencies.

---

[2]In this paper, **early** refers to time steps closer to $t = 0$, while **recent** refers to those closer to $t = T$.

$$MReLU: \quad \mathbf{I} = ReLU(\mathbf{X}) - \mathbf{X} \tag{6}$$

$$V_{\text{th}}[t] = V_{\text{th}}[t-1] + \gamma \cdot \tanh(I[t-1]) \tag{7}$$

Specifically, as Fig. (1) (**a**), we implement this design through a customized Mask-ReLU (MReLU) layer. Denote the inhibitory signal sequence as $\mathbf{I} = [I[0], I[1], \cdots, I[T]]$, and we obtain $\mathbf{I}$ from the input sequence $\mathbf{X}$ using Eq. (6). Furthermore, across the temporal dimension, the membrane potential threshold $V_{\text{th}}$ gradually accumulates the inhibitory signal $\mathbf{I}$ as Eq. (7), where $\tanh$ represents the hyperbolic tangent function, and $\gamma$ is a constant coefficient set to a default value of 0.1.

We analyze why LLIF neurons enhance long-term dependencies from the perspective of gradient contribution in Appendix C. To achieve stronger long-term dependencies, the contribution from early time steps must have a higher expected proportion in the total gradient compared to standard LIF neurons. This can alleviate the inherent gradient decay phenomenon in BPTT caused by Eq. (4), thereby helping the network more effectively retain and leverage information from distant time steps during the backpropagation process.

**Short-LIF (SLIF).** Strengthening short-term dependency requires neurons to enhance their sensitivity to recent time steps. Based on the synaptic potentiation phenomenon (Zucker & Regehr, 2002) observed in biological neurons—where a neuron temporarily increases its sensitivity to recent signals when subjected to strong stimulation while moderately reducing the influence of historical activations—the design in SLIF neurons contrasts with the inhibitory signal design in LLIF. In SLIF, we treat membrane potentials exceeding the threshold as signals of over-excitation, linking this signal to the sensitivity of signal reception in future time steps. This implies that in SLIF neurons, the rate of decay for past accumulated potentials at the current time step is positively correlated with the intensity of recent stimulation, enabling the neuron to pay more attention to recent inputs. Specifically, in SLIF, we employ a soft reset method (detailed in Appendix A), where the degree of recent stimulation is represented by the residual membrane potential, which refers to the neuron's potential after a soft reset. The decay rate of the potential is controlled by $\tau$. Thus, the SLIF process, as depicted in Fig. (1) (**b**), defines the residual membrane potential $V_p$ by Eq. (8), while the dynamic between the membrane time constant $\tau$ and the residual membrane potential $V_p$ is given by Eq. (9).

$$V_p[t] = S[t] \cdot (V[t] - V_{\text{th}}) \tag{8}$$

$$\tau[t] = \max[1.05, \tau_0 - \beta \cdot \tanh(V_p[t-1])] \tag{9}$$

For Eq. (9), $\tau_0$ is the membrane time constant at initialization, $\tanh$ is the hyperbolic tangent function, and $\beta$ is a constant coefficient used to regulate the amplitude, with a default value set to 0.1. Additionally, based on the findings of Fang et al., 2021b and Deng et al., 2022, 1 is considered the lower bound for $\tau$. Therefore, we set a slightly larger minimum value of $\tau$ at 1.05 to ensure stability and effectiveness in the model's behavior.

From the perspective of backpropagation, we can conclude that when SLIF adapts the membrane time constant using Eq. (9), it increases the contribution of recent time steps to the overall gradient compared to standard LIF (see Appendix D for a detailed derivation).

## 4.2 ABLATION STUDY: WHY A NEURON SEARCH STRATEGY IS ESSENTIAL?

Although it may seem intuitive to optimize the network by placing SLIF neurons in shallow layers and LLIF neurons in deep layers based on the dynamics of dependency and neuron design discussed earlier, the reality is far more complex. Due to the intricate connection structures in modern network architectures, such as skip connections (He et al., 2016), the temporal dependency varies non-linearly along the spatial dimension. Moreover, the spiking neuron layers' requirements for temporal dependency differ depending on the task type—for example, static datasets versus neuromorphic datasets have inherently distinct needs, which is intuitive but challenging to address manually. In this section, we demonstrate through ablation experiments that manual placement of neurons is suboptimal, which motivates the development of our neuron search strategy.

The core idea of this ablation study is to replace the LIF neurons at different depths in a conventional LIF-based network with our dependency-enhanced neurons. By observing the performance changes resulting from neuron replacement at various depths, we aim to demonstrate the differing responses to neuron types across network depths. For this study, we selected two representative datasets and corresponding networks: the static dataset CIFAR-10 using ResNet-18 and the neuromorphic dataset DVS-CIFAR-10 using VGG-11. ResNet-18 consists of 8 residual blocks, each containing two spiking neuron layers, while VGG-11 has 5 blocks, with the sequence of spiking neuron layers across the blocks being 1, 1, 2, 2, and 2 layers, respectively, from the input to the output layer. Based on these architectures, we define the methods for this ablation experiment as follows:

- **LIF / LLIF / SLIF.** LIF / LLIF / SLIF neurons for all spiking neuron layers.

- **LLIF-$X$ / SLIF-$X$.** (Larger $X$ corresponding to deeper layers)

    - In ResNet-18: LLIF / SLIF neurons are applied to the spiking neuron layers from the $(2X-1)$-th to the $(2X)$-th blocks, while LIF neurons are used for others.
    - In VGG-11: LLIF / SLIF neurons are applied to the spiking neuron layers in the $X$-th block, while LIF neurons are used for others.

For all methods within each dataset, we maintained identical experimental environments and hyperparameter settings. Detailed experimental setups are provided in Appendix F. The comparative results for each method are summarized in Table 1. Additionally, due to the large number of methods evaluated in this experiment, we selected four representative methods from both the CIFAR-10 and DVS-CIFAR-10 datasets—LIF, LLIF-4, SLIF, and SLIF-4—to visualize test accuracy and loss over iterations, as shown in Fig. 2. The full test accuracy curves for all methods are available in Appendix G. It's important to note that methods with the same name, such as SLIF-1, have different implications when applied to ResNet-18 and VGG-11.

Table 1: Comparison of LIF variants

| Method | Test Accuracy (%) | |
| --- | --- | --- |
| | **CIFAR-10** | **DVS-CIFAR-10** |
| | ResNet-18 | VGG-11 |
| LIF | 93.46 | 76.20 |
| LLIF | 94.78 | 77.00 |
| LLIF-1 | 94.36 | 77.00 |
| LLIF-2 | 94.71 | 76.60 |
| LLIF-3 | 94.70 | 77.10 |
| LLIF-4 | **94.83** | **79.30** |
| LLIF-5 | / | 77.80 |
| SLIF | 83.83 | 66.80 |
| SLIF-1 | 94.41 | 77.50 |
| SLIF-2 | 93.30 | 76.20 |
| SLIF-3 | 92.03 | 76.10 |
| SLIF-4 | 90.28 | 78.00 |
| SLIF-5 | / | 76.20 |

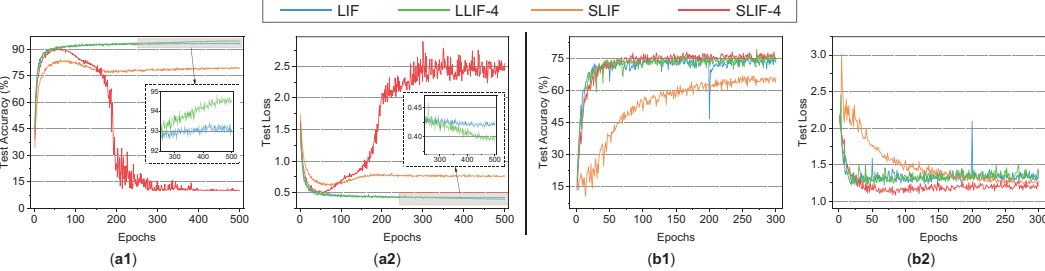

Figure 2: Comparison of four representative methods. Where (**a1**) and (**a2**) are the test accuracy and loss iteration performance of methods based on ResNet-18 on CIFAR-10, respectively. Similarly, (**b1**) and (**b2**) are the corresponding performance of methods based on VGG-11 on DVS-CIFAR-10.

First, focusing on CIFAR-10, as shown in Table 1, it is clear that all methods based on LLIF demonstrate significant performance improvements compared to standard LIF, with only minor differences among them. Notably, LLIF-4 achieved the highest test accuracy among these methods. In contrast, for SLIF-based methods, only SLIF-1 outperforms standard LIF, while SLIF and SLIF-2 through SLIF-4 show some performance degradation. Particularly, SLIF achieves only 83.83% accuracy. As illustrated in Fig. 2 (**a1**) and (**a2**), both SLIF and SLIF-4 exhibit severe overfitting before epoch

100, with SLIF showing a more pronounced decline in accuracy. By the end of 300 epochs, SLIF-4's performance is nearly equivalent to its untrained state. This is because the closer the network is to the output layer, the stronger its temporal dynamics become, and the more abstract its temporal features. As such, near-output neurons need to integrate contextual information through long-term dependencies. If these neurons are forced to emphasize short-term dependencies, as seen with SLIF-4, it can severely damage network performance. On the other hand, on the DVS-CIFAR-10 dataset, LLIF-4 again achieved the highest accuracy, and LLIF-based methods exhibit similar trends as those observed on CIFAR-10. As shown in Fig. 2 (**b1**) and (**b2**), SLIF-based methods do not exhibit over-fitting; however, except for SLIF, their performance is comparable to that of standard LIF. Notably, SLIF shows very slow convergence, achieving only 66.80% accuracy.

The ablation experiments above validate our analysis: enhancing short-term dependency in shallow layers (e.g., SLIF-1) and long-term dependency in deep layers (e.g., LLIF-4) can improve network performance. However, these experiments also lead to two important conclusions. First, within the same dataset, the placement of different neuron types has a significant impact on network performance. Second, the performance trends resulting from neuron placement along the network's spatial dimension vary across different datasets.

Based on these conclusions, while manually replacing shallow neurons with SLIF and deep neurons with LLIF can improve performance to some extent, this approach is inherently limited in its ability to fully explore the network's maximum potential. The reason is that there are numerous possible combinations of LIF, LLIF, and SLIF neurons along the spatial dimension, and manual placement is unlikely to yield optimal results. Thus, an adaptive method is required—one that can automatically search for the optimal neuron placement throughout the entire network architecture. This approach would release the potential of design neurons, optimizing both model performance and stability.

## 4.3 NEURON SEARCH

In this section, we introduce the Layer-Wise Neuron Search (LWNS) strategy, built on the foundation of Differentiable Architecture Search (DARTS) (Liu et al., 2018), to enable the spatial adaptation of temporal dependencies in SNNs. DARTS is a neural architecture search method that efficiently optimizes network architectures by transforming the discrete search space into a continuous one, facilitating gradient-based optimization. This approach allows us to dynamically search for the optimal neuron placement across different layers, enhancing the network's ability to adapt to varying temporal dependencies along the spatial dimension.

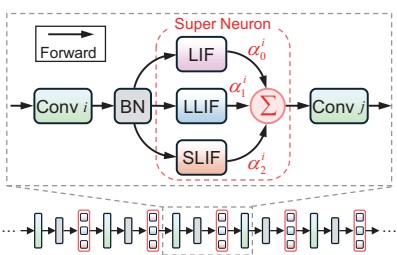

Figure 3: Forward pass of LWNS-Net.

Unlike traditional DARTS, which perform architecture search based on cell structures, LWNS retains the classic architectures of ResNet or VGG and conducts a layer-wise search across the entire network. Specifically, in the LWNS network (LWNS-Net), all spiking neurons are replaced with super neurons, each composed of three types of LIF neurons. This allows LWNS to use gradient descent to select the most suitable neuron type for each layer, thereby adapting flexibly to the temporal dependency requirements of different layers. This method not only extends the applicability of DARTS but also allows for more flexible and refined neuron configuration at the layer level.

The construction of the LWNS-Net and the super neurons is depicted in Fig. 3. When data flows from convolution layer $i$ to convolution layer $j$ (where $j = i + 1$), it first passes through a normalization layer for preprocessing, followed by activation through the super neuron. Within the super neuron, the inputs to LIF, LLIF, and SLIF neurons are identical. Their outputs are denoted as $O_0^i$, $O_1^i$, and $O_2^i$, respectively. The input to convolution layer $j$ is given by Eq. (10), where $\alpha$ represents the architecture parameters, which are learned using the same approximated gradient methods as described in Liu et al., 2018. Additionally, the size of search space of LWNS depends on the number of spiking neuron layers in the specific network framework. Assuming there are $N$ spiking neuron layers in the network, the search space size of LWNS would be $3^N$, which is compact, reducing search cost and enhancing network performance with minimal additional training resources.

$$IN_j = \sum_{k=0}^{2} \frac{\exp(\alpha_k^i)}{\sum_{l=0}^{2} \exp(\alpha_l^i)} \mathrm{BN}(O_k^i) \qquad (10)$$

Table 2: Comparison with SOTA on CIFAR-10, CIFAR-100, Tiny-ImageNet, DVS-CIFAR-10, and DVS-Gesture datasets.

| | Model | Method | Architecture | Time step | Accuracy (%) |
|---|---|---|---|---|---|
| CIFAR-10 | Offline LTL (Yang et al., 2022) | A2S | ResNet-20 | 32 | 95.28 |
| | Joint A-SNN (Guo et al., 2023a) | A2S | ResNet-18 | 4 | 95.45 |
| | TET-SSF (Wang et al., 2023a) | SG | ResNet-18 | 20 | 94.90 |
| | OTTT (Xiao et al., 2022) | SG | VGG-11$^{\ddagger}$ | 6 | 93.73 |
| | SLTT (Meng et al., 2023) | SG | ResNet-18 | 6 | 94.59 |
| | LT-SNN (Hasssan et al., 2024) | SG | Spikformer | 4 | 95.19 |
| | DTA-TTFS (Wei et al., 2023) | SG | VGG-16 | 8 | 93.05 |
| | ESL-SNNs (Shen et al., 2023) | SG | ResNet-19 | 4 | 91.09 |
| | SNASNet-Bw (Kim et al., 2022) | SG | NAS | 8 | 94.37 |
| | AutoSNN (Na et al., 2022) | SG | NAS | 8 | 93.15 |
| | **ST-DANO (Ours)** | SG | ResNet-18$^{*}$ | 8 / 6 / 4 / 2 | **95.92 / 95.65 / 95.66 / 95.50** |
| CIFAR-100 | Offline LTL (Yang et al., 2022) | A2S | ResNet-20 | 32 | 76.12 |
| | Joint A-SNN (Guo et al., 2023a) | A2S | ResNet-18 | 4 | **77.39** |
| | RecDis-SSF (Wang et al., 2023a) | SG | ResNet-18 | 20 | 75.48 |
| | OTTT (Xiao et al., 2022) | SG | VGG-11$^{\ddagger}$ | 6 | 71.11 |
| | sparse-KD (Xu et al., 2024b) | SG | ResNet-18 | 4 | 73.01 |
| | SLTT (Meng et al., 2023) | SG | ResNet-18 | 6 | 74.67 |
| | LT-SNN (Hasssan et al., 2024) | SG | ResNet-19 | 6 | 74.82 |
| | DTA-TTFS (Wei et al., 2023) | SG | VGG-16 | 8 | 69.66 |
| | ESL-SNNs (Shen et al., 2023) | SG | ResNet-19 | 4 | 73.48 |
| | SNASNet-Bw (Kim et al., 2022) | SG | NAS | 5 | 73.40 |
| | AutoSNN (Na et al., 2022) | SG | NAS | 8 | 69.16 |
| | **ST-DANO (Ours)** | SG | ResNet-18$^{*}$ | 8 / 6 / 4 / 2 | 76.25 / 75.94 / 75.93 / 75.53 |
| Tiny-ImageNet | Offline LTL (Yang et al., 2022) | A2S | ResNet-20 | 32 | 57.73 |
| | Joint A-SNN (Guo et al., 2023a) | A2S | VGG-16 | 4 | 55.39 |
| | SEW-SSF (Wang et al., 2023a) | SG | ResNet-34 | 20 | 58.81 |
| | default-KD (Xu et al., 2024b) | SG | ResNet-18 | 4 | 59.31 |
| | SNASNet-Bw (Kim et al., 2022) | SG | NAS | 5 | 55.08 |
| | AutoSNN (Na et al., 2022) | SG | NAS | 8 | 46.79 |
| | **ST-DANO (Ours)** | SG | VGG-13$^{*}$ | 8 / 6 / 4 / 2 | **60.24 / 60.14 / 60.08** / 59.10 |
| DVS-CIFAR-10 | CE-SSF (Wang et al., 2023a) | SG | VGG-11 | 20 | 78.00 |
| | OTTT (Xiao et al., 2022) | SG | VGG-11$^{\ddagger}$ | 10 | 77.10 |
| | SLTT (Meng et al., 2023) | SG | ResNet-18 | 10 | 77.30 |
| | LT-SNN (Hasssan et al., 2024) | SG | VGG-7 | 10 | 80.20 |
| | ESL-SNNs (Shen et al., 2023) | SG | VGG-9 | 10 | 78.30 |
| | AutoSNN (Na et al., 2022) | SG | NAS | 20 | 72.50 |
| | **ST-DANO / ST-DANO$^{\dagger}$ (Ours)** | SG | VGG-11$^{*}$ | 10 | **79.30 / 83.90** |
| | **ST-DANO / ST-DANO$^{\dagger}$ (Ours)** | SG | VGG-11$^{*}$ | 8 | 77.20 / 81.80 |
| DVS-Gesture | OTTT (Xiao et al., 2022) | SG | VGG-11$^{\ddagger}$ | 20 | 96.88 |
| | sparse-KD (Xu et al., 2024b) | SG | 5Conv 1FC | 16 | 96.18 |
| | SLTT (Meng et al., 2023) | SG | VGG-11 | 20 | **98.62** |
| | AutoSNN (Na et al., 2022) | SG | NAS | 20 | 96.53 |
| | **ST-DANO (Ours)** | SG | VGG-11$^{*}$ | 20 / 16 / 8 | 98.61 / 97.22 / 96.88 |

$^{\dagger}$ Data augmentation (Li et al., 2022), $^{\ddagger}$ Scaled weight standardization (Qiao et al., 2019),
$^{*}$ Optimal neuron configuration obtained by ST-DANO.

## 5 EXPERIMENTS

### 5.1 COMPARISON WITH STATE-OF-THE-ART APPROACHES

To demonstrate the effectiveness of ST-DANO in enhancing the performance of SNNs, we conduct experiments on static datasets CIFAR-10 (Krizhevsky et al., 2009), CIFAR-100 (Krizhevsky et al., 2009), and Tiny-ImageNet (Deng et al., 2009), as well as neuromorphic datasets DVS-CIFAR-10 (Li et al., 2017) and DVS-Gesture (Amir et al., 2017). The results are then analyzed and compared against the state-of-the-art methods. Detailed descriptions and process methods of the datasets are provided in Appendix E. The training process of ST-DANO across all datasets is divided into two phases: searching and retraining. In the first phase, the optimal neuron configuration is searched based on ST-DANO, while in the later one, the network with optimal neuron configuration will be retrained. The complete and detailed implementation of both phases is provided in Appendix F. Additionally, to thoroughly evaluate the performance of ST-DANO, we search for the optimal neuron configurations corresponding to different time-step settings across participated datasets.

**Static Datasets.** As shown in Table 2, ST-DANO achieves the best performance on the CIFAR-10 and Tiny-ImageNet datasets among the three static datasets, with accuracies of 95.92% and 60.24%, respectively. Although the Joint A-SNN (Guo et al., 2023a) outperforms in CIFAR-100 with an accuracy of 77.39%, our method still surpasses other surrogate gradient-based learning methods in this dataset. An interesting observation is that on CIFAR-10, our method performs slightly better with a time step setting of 4 compared to that of 6. This suggests that when ST-DANO searches for the neuron configuration on the CIFAR-10 dataset with a time step setting of 6, it finds a suboptimal rather than the desired optimal configuration. Despite this, our method still significantly outperforms other approaches, underscoring its tremendous advantages.

**Neuromorphic Datasets.** On the DVS-CIFAR-10 dataset, our method achieves the best accuracy of 79.30% without data augmentation, and further improves to 83.90% with augmentation applied. On DVS-Gesture, ST-DANO achieves an accuracy of 98.61%, only slightly below that of SLTT (Meng et al., 2023) (98.62%). Overall, our method demonstrates equally strong performance on challenging neuromorphic datasets as it does on static datasets.

### 5.2 OPTIMAL NEURON CONFIGURATIONS

Table 3 provides the optimal neuron configurations obtained by ST-DANO and the search cost, with the corresponding architecture parameters visualized in Appendix H. Notably, on the CIFAR-10 and CIFAR-100 datasets, the first and fifth layers of the shallow network tend to use SLIF, while the deeper layers, especially the last four, prefer LLIF. This observation aligns with our original analysis in designing ST-DANO: shallow layers prefer short-term dependencies, while deep layers prefer long-term dependencies. Meanwhile, it also confirms that the network's temporal dependency requirements are not linear. For example, in the optimal neuron configurations of CIFAR-10 and CIFAR-100, shallow layers 2, 3, and 4 do not favor LLIF but instead use standard LIF. For neuromorphic datasets, they predominantly favor long-term dependency neurons. This is because, unlike static datasets, the input data in neuromorphic datasets is inherently dynamic and discrete time-series data, requiring the overall network to integrate information over longer time ranges. Interestingly, the optimal neuron configuration for DVS-CIFAR-10 with a time step of 10 is coincidentally identical to the LLIF-4 in VGG-11 we manually set in Section 4.2.

From the heatmaps of the architecture parameters provided in Table 3, it can be observed that on static datasets, as the network deepens, the architecture parameters of SLIF neurons show a downward trend, while those of LLIF neurons increase. However, on neuromorphic datasets such as DVG-Gesture, LLIF neurons dominate across all layers of the network. These findings are consistent with our previous analysis of temporal dependencies.

Furthermore, in Appendix I, we provide a comparison of the spike firing rates during the inference phase between the optimal neuron configuration found by our method and the conventional LIF network across all cases. The results demonstrate that our approach not only improves network performance but also effectively suppresses spike activity, thereby reducing inference energy consumption, making it more favorable for real-world deployment.

Table 3: Optimal neuron configurations discovered by ST-DANO across different datasets. The sequence of numbers in the '**Optimal Neuron Configuration**' represents the types of neurons used from the input layer to the output layer of the network, where 0 denotes LIF neurons, 1 denotes LLIF neurons, and 2 denotes SLIF neurons. '**Search Cost**' is measured in GPU days. '**Architecture Parameter Visualization**' representatively displays heatmaps of the architecture parameters under time step settings of 6, 6, 6, 10, and 20 on each dataset. Each subfigure consists of three rows and $N$ columns, where the first, second, and third rows correspond to SLIF, LLIF, and LIF neurons, respectively, and $N$ represents the number of spiking neuron layers. In the heatmaps, the redder the color, the larger the architecture parameter, while the closer it is to blue, the smaller the value. Full edition of architecture parameter visualization is provided in Appendix H.

| | Arc. | Time Step | Optimal Neuron Configuration | Search Cost | Architecture Parameter Visualization |
|---|---|---|---|---|---|
| CIFAR-10 | ResNet-18 | 2 | 20002000000001001 | 0.13 | |
| | | 4 | 20002000000001001 | 0.16 | |
| | | 6 | 20000000000001011 | 0.19 | |
| | | 8 | 20000000000001011 | 0.22 | |
| CIFAR-100 | ResNet-18 | 2 | 20002000000010011 | 0.13 | |
| | | 4 | 20002000000010111 | 0.16 | |
| | | 6 | 20002000000010111 | 0.19 | |
| | | 8 | 20002000001010111 | 0.22 | |
| Tiny-ImageNet | VGG-13 | 2 | 000020101 | 0.37 | |
| | | 4 | 000000101 | 0.48 | |
| | | 6 | 000000101 | 0.59 | |
| | | 8 | 000020101 | 0.70 | |
| DVS-CIFAR-10 | VGG-11 | 8 | 00011100 | 0.11 | |
| | | 10 | 00001100 | 0.13 | |
| DVS-Gesture | VGG-11 | 8 | 11000001 | 0.10 | |
| | | 16 | 10112111 | 0.17 | |
| | | 20 | 10111111 | 0.21 | |

Besides, beyond the commonly used architectures in Table 2, we conduct **additional experiments** using ST-DANO. The results of these additional experiments further demonstrate the effectiveness of our method and reinforce the conclusions above. For detailed information on the training settings, experimental results, and the optimal neuron configurations discovered during these additional experiments, please refer to Appendix J.

## 6 CONCLUSION

The BPTT method has streamlined the training of SNNs, while direct input encoding dramatically cuts down the number of time steps needed. Together, these two approaches have drastically enhanced training efficiency and effectiveness, establishing themselves as the dominant framework in SNN research. However, the variation in neurons' temporal dependency requirements across the spatial dimension, introduced by this combination, poses a significant barrier to unlocking the full performance potential of SNNs. To address this issue, we propose a Spatio-Temporal Dependency-Aware Neuron Optimization (ST-DANO) method to find the optimal dependency configuration for the network architecture. Specifically, we first design two variants of LIF neurons: LLIF and SLIF, which are capable of enhancing neurons' ability to capture long-term and short-term dependencies, respectively. Building on this, we design a layer-wise neuron search strategy based on DARTS, which utilizes a differentiable search process to find the optimal temporal dependency configuration for each spiking neuron layer, enabling adaptive temporal dependency along the spatial dimension in SNNs. Finally, extensive experiments on static and neuromorphic datasets demonstrate the superiority of our method.

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

## APPENDIX

## A RESET METHODS AND SURROGATE GRADIENT

In LIF neurons, the membrane potential undergoes a reset following the generation of an action potential (spike). There are generally two forms of resetting: hard reset (Deng et al., 2022; Zhu et al., 2024) and soft reset (Han et al., 2020; Park et al., 2024). At time step $t$, the two reset processes can be described as follows.

$$V[t] \longleftarrow \begin{cases} (1 - S[t]) \cdot V[t] & \text{Hard Reset} \\ V[t] - S[t] \cdot V_{\text{th}} & \text{Soft Reset} \end{cases} \tag{11}$$

For all types of spiking neurons, we use the piecewise quadratic function (triangle function) as the surrogate gradient (Fang et al., 2023; Neftci et al., 2019), which is defined as:

$$g'(x) = \begin{cases} 0, & |x| > \frac{1}{\alpha} \\ -\alpha^2 |x| + \alpha, & |x| \le \frac{1}{\alpha} \end{cases} \tag{12}$$

The primitive function is defined as:

$$g(x) = \begin{cases} 0, & x < -\frac{1}{\alpha} \\ -\frac{1}{2}\alpha^2 |x| x + \alpha x + \frac{1}{2}, & |x| \le \frac{1}{\alpha} \\ 1, & x > \frac{1}{\alpha} \end{cases} \tag{13}$$

where constant $\alpha$ is set to 1 for all cases.

## B GRADIENT OF LIF-BASED BPTT

In this section, we infer the gradient over the temporal dimension for LIF neurons during the BPTT process.

The following is the dynamic equation of LIF neurons in hard reset mode:

$$V[t] = V[t-1] \cdot \exp(-\frac{1}{\tau}) \cdot (1 - \Theta(V[t-1] - V_{\text{th}})) + X[t] \tag{14}$$

The core of BPTT is to perform backpropagation after all time steps have been completed, and accumulate the gradients over all time steps. Assuming the total number of time steps is $T$, for any time step $t \in [0, 1, \ldots, T]$, the neuron output $S[t]$ at this time step is related to the inputs at all

previous time steps $X[0], X[1], \ldots, X[t]$. Therefore, the gradient at time step $t$, $\nabla_{X[0:t]} S[t]$, follows the relationship:

$$\nabla_{\mathbf{X}[0:t]} S[t] = \sum_{i=0}^{t} \frac{\partial S[t]}{\partial X[i]} = \frac{\partial S[t]}{\partial X[0]} + \frac{\partial S[t]}{\partial X[1]} + \cdots + \frac{\partial S[t]}{\partial X[t]} \tag{15}$$

where $\mathbf{X}[0:t] = [X[0], X[1], \cdots, X[t]]$, and therefore:

$$\nabla_{\mathbf{X}} \mathbf{S} = \sum_{i=0}^{T} \nabla_{\mathbf{X}[0:i]} S[i] = \sum_{t=0}^{T} \sum_{i=0}^{t} \frac{\partial S[t]}{\partial X[i]} \tag{16}$$

For the term $\sum_{i=0}^{t} \frac{\partial S[t]}{\partial X[i]}$, it can be expanded as $\sum_{i=0}^{t} \frac{\partial S[t]}{\partial V[i]} \frac{\partial V[i]}{\partial X[i]}$. Using Eq. (14), we have $\frac{\partial V[i]}{\partial X[i]} = 1$. Therefore, Eq. (16) can be rewritten as:

$$\nabla_{\mathbf{X}} \mathbf{S} = \sum_{t=0}^{T} \sum_{i=0}^{t} \frac{\partial S[t]}{\partial V[i]} \tag{17}$$

Thus, we will now rewrite Eq. (15) as Eq. (18) and expand each term on the right-hand side in detail:

$$\nabla_{\mathbf{X}[0:t]} S[t] = \sum_{i=0}^{t} \frac{\partial S[t]}{\partial V[i]} = \frac{\partial S[t]}{\partial V[0]} + \frac{\partial S[t]}{\partial V[1]} + \cdots + \frac{\partial S[t]}{\partial V[t]} \tag{18}$$

where:
$$\frac{\partial S[t]}{\partial V[t]} = \Theta'(V[t] - V_{\text{th}}) \tag{19}$$

$$\frac{\partial S[t]}{\partial V[t-1]} = \frac{\partial S[t]}{\partial V[t]} \frac{\partial V[t]}{\partial V[t-1]}$$
$$= \Theta'(V[t] - V_{\text{th}}) \cdot \exp(-\frac{1}{\tau}) \cdot (1 - \Theta(v[t-1] - V_{\text{th}}) - V[t-1] \cdot \Theta'(V[t-1] - V_{\text{th}})) \tag{20}$$

$$\frac{\partial S[t]}{\partial V[t-2]} = \frac{\partial S[t]}{\partial V[t]} \frac{\partial V[t]}{\partial V[t-1]} \frac{\partial V[t-1]}{\partial V[t-2]}$$
$$= \Theta'(V[t] - V_{\text{th}}) \cdot \exp(-\frac{2}{\tau}) \cdot \prod_{i=1}^{2} (1 - \Theta(v[t-i] - V_{\text{th}}) - V[t-i] \cdot \Theta'(V[t-i] - V_{\text{th}})) \tag{21}$$

By partially expanding Eq. (18) through Eqs. (19)-(21), we can summarize the gradient at time step $t$ as:

$$\nabla_{\mathbf{X}[0:t]} S[t] = \Theta'(V[t] - V_{\text{th}}) \cdot (1 + \exp(-\frac{1}{\tau}) \cdot \kappa_h(V[t-1]) + \exp(-\frac{2}{\tau}) \cdot \kappa_h(V[t-1]) \cdot \kappa_h(V[t-2]) + \cdots)$$
$$= \Theta'(V[t] - V_{\text{th}}) \cdot (1 + \sum_{i=1}^{t} \exp(-\frac{i}{\tau}) \cdot \prod_{j=1}^{i} \kappa_h(V[t-j])) \tag{22}$$

where

$$\kappa_h(V[x]) = 1 - \Theta(V[x] - V_{\text{th}}) - V[x] \cdot \Theta'(V[x] - V_{\text{th}}) \tag{23}$$

Since the dynamics of the soft reset follow Eq. (24), if the neuron uses soft reset, the gradient calculation only requires replacing $\kappa_h$ in Eq. (22) with $\kappa_s$ from Eq. (25).

$$V[t] = (V[t-1] - \Theta(V[t-1] - V_{\text{th}}) \cdot V_{\text{th}}) \cdot \exp(-\frac{1}{\tau}) + X[t] \tag{24}$$

$$\kappa_s(V[x]) = 1 - V_{\text{th}} \cdot \Theta'(V[x] - V_{\text{th}}) \tag{25}$$

In summary, we can derive the gradient of the neuron during a single backpropagation in BPTT (with a total of $T$ time steps) as Eq. (C), where the term $\Theta'(V[t] - V_{\text{th}})$ can be replaced by Eq. (12).

$$\nabla_{\mathbf{X}}\mathbf{S} = \sum_{t=0}^{T} \Theta'(V[t] - V_{\text{th}}) \cdot (1 + \sum_{i=0}^{t} \exp(-\frac{i}{\tau}) \cdot \prod_{j=1}^{i} \kappa_h(V[t-j])) \tag{26}$$

## C    ANALYSIS OF IMPACT OF $V_{th}$ IN LLIF'S GRADIENT

In this section, we analyze the impact of $V_{th}$ variation on the contribution of gradient components at different time steps to the total gradient. According to Equation , the terms related to $V_{th}$ include $\theta'(V[t] - V_{th})$, the $\theta(V[t] - V_{th})$ term in $\kappa_h$, and $V[x] \cdot \theta'(V[t] - V_{th})$. For the first term $\theta'(V[t] - V_{th})$, due to the properties of the $\Theta$ function being invariant across time steps, this term does not affect the contribution of gradient components at different time steps to the total gradient from an expectation perspective. Based on this analysis, we construct a contribution function $G(V_{th})$ with $T$ terms, as shown in Eq. (27), to facilitate our analysis. Notably, the first term in Eq. (27), denoted as $G_T$, represents the expected contribution of the $T$-th time step to the total gradient. Correspondingly, the last term, denoted as $G_1$, represents the expected contribution of the first time step to the total gradient.

$$G(V_{th}) = \underbrace{\exp\left(-\frac{1}{\tau}\right) \cdot \kappa_h(V[T])}_{G_T} + \underbrace{[\exp\left(-\frac{1}{\tau}\right) \cdot \kappa_h(V[T]) + \exp\left(-\frac{2}{\tau}\right) \cdot \kappa_h(V[T]) \cdot \kappa_h(V[T-1])]}_{G_{T-1}}$$

$$+ \cdots + \underbrace{[\exp\left(-\frac{1}{\tau}\right) \cdot \kappa_h(V[T]) + \exp\left(-\frac{2}{\tau}\right) \cdot \kappa_h(V[T]) \cdot \kappa_h(V[T-1]) + \cdots]}_{G_1}$$

$$= \sum_{t=0}^{T} \sum_{i=0}^{t} \exp\left(-\frac{i}{\tau}\right) \cdot \prod_{j=1}^{i} \kappa_h(V[t-j]) \tag{27}$$

Based on $G(V_{th})$, we construct a proportional distribution sequence $\mathbf{P}$, as shown in Eq. (28), to represent the expected contribution proportion of each time step to the total gradient. In this sequence, each term expresses the proportion of the total gradient attributed to the corresponding time step.

$$\mathbf{P} = [\frac{G_T}{G(V_{th})}, \frac{G_{T-1}}{G(V_{th})}, \cdots, \frac{G_1}{G(V_{th})}] \tag{28}$$

If $V_{th}$ changes, we denote the updated threshold value as $V'_{th}$. We can then construct another proportional distribution sequence $\mathbf{P}'$ as shown in Eq. (29), which represents the proportion of each gradient component in the total amount within $G(V'_{th})$.

$$\mathbf{P}' = [\frac{G'_T}{G(V'_{th})}, \frac{G'_{T-1}}{G(V'_{th})}, \cdots, \frac{G'_1}{G(V'_{th})}] \tag{29}$$

At this point, we compute the difference between $\mathbf{P}'$ and $\mathbf{P}$, resulting in the change sequence $\Delta\mathbf{P}$, as shown in Eq. (30). $\Delta\mathbf{P}$ represents the change in the contribution proportion of each time step to the gradient after the adaptive adjustment of $V_{th}$. Both $\mathbf{P}$ and $\mathbf{P}'$ are based on the function $G(V_{th})$, which is expectationally an increasing sequence, thus, to demonstrate that the gradient contribution proportion of recent time steps increases while that of early time steps decreases, we only need to prove that the $\Delta\mathbf{P}$ sequence is an increasing sequence. This would confirm that adaptive $V_{th}$ strengthens the neuron's long-term dependency.

$$\Delta\mathbf{P} = \mathbf{P}' - \mathbf{P} = [\underbrace{\frac{G'_T}{G(V'_{th})} - \frac{G_T}{G(V_{th})}}_{\Delta P_T}, \underbrace{\frac{G'_{T-1}}{G(V'_{th})} - \frac{G_{T-1}}{G(V_{th})}}_{\Delta P_{T-1}}, \cdots, \underbrace{\frac{G'_1}{G(V'_{th})} - \frac{G_1}{G(V_{th})}}_{\Delta P_1}] \tag{30}$$

To prove that the sequence $\Delta\mathbf{P}$ is an increasing sequence, we only need to show that for any terms $\Delta P_i$ and $\Delta P_{i-1}$ within $\Delta\mathbf{P}$, the relation $\Delta P_i - \Delta P_{i-1} < 0$ holds. Subtracting these two terms, we derive the following relationship:

$$\begin{aligned} \Delta P_i - \Delta P_{i-1} &= \frac{G'_i}{G(V'_{th})} - \frac{G_i}{G(V_{th})} - \left(\frac{G'_{i-1}}{G(V'_{th})} - \frac{G_{i-1}}{G(V_{th})}\right) \\ &= \frac{G'_i - G'_{i-1}}{G(V'_{th})} - \frac{G_i - G_{i-1}}{G(V_{th})} \end{aligned} \tag{31}$$

Therefore, we construct a function as shown in Eq. (32) and proceed to discuss its monotonicity.

$$F(x) = \frac{G_i(x) - G_{i-1}(x)}{x} \tag{32}$$

Based on Eq. (27), we can derive the relationship among the different contributing components as follows:

$$G_i(x) - G_{i-1}(x) = \exp\left(-\frac{T+2-i}{\tau}\right) \cdot \prod_{m=i-1}^{T} \kappa_h(V[m], x) \tag{33}$$

where
$$\kappa_h(V[m], x) = 1 - \Theta(V[m] - x) - V[m] \cdot \Theta'(V[m] - x) \tag{34}$$

Here, in the subsequent analysis, we replace $\Theta'$ in Eq. (34) using surrogate gradients, as shown in Eq. (12).

Substituting Eq. (33) into Eq. (32), we transform it into the following relationship:

$$F(x) = \frac{N(x)}{G(x)} = \frac{\exp\left(-\frac{T+2-i}{\tau}\right) \cdot \prod_{m=i-1}^{T} \kappa'_h(V[m], x)}{G(x)} \tag{35}$$

To ensure that Eq. (31) holds when $V'_{th} > V_{th}$, we need to prove that the expectation of $F(x)$ is monotonically decreasing when $x > 0$.

First, for the $\kappa_h$ function in $F(x)$, we conduct a case-by-case analysis. Based on the value of $V[m]-x$, we can decompose and separately define $\kappa_h(V[m]-x)$ in piecewise fashion. $g'(V[m]-x)$ can be divided into two cases depending on the value of $|V[m] - x|$:

$$g'(V[m] - x) = \begin{cases} 0, & |V[m] - x| > \frac{1}{\alpha} \\ -\alpha^2|V[m] - x| + \alpha, & |V[m] - x| \le \frac{1}{\alpha} \end{cases} \tag{36}$$

Based on the above equation, we can provide a piecewise definition for $\kappa_h(V[m] - x)$ as follows:

$$\kappa_h(V[m] - x) = \begin{cases} 1, & V[m] - x \le -\frac{1}{\alpha} \\ 1 - \alpha^2(V[m] - x) + \alpha, & -\frac{1}{\alpha} < V[m] - x \le 0 \\ \alpha^2(V[m] - x) + \alpha, & 0 < V[m] - x \le \frac{1}{\alpha} \\ 0, & V[m] - x > \frac{1}{\alpha} \end{cases} \qquad (37)$$

Taking the derivative of $F(x)$, we obtain:

$$\frac{d}{dx}F(x) = \frac{d}{dx}\left(\frac{N(x)}{G(x)}\right) = \frac{N'(x)G(x) - N(x)G'(x)}{[G(x)]^2} = F(x) \cdot \left(\frac{N'(x)}{N(x)} - \frac{G'(x)}{G(x)}\right) \qquad (38)$$

Next, we separately analyze the functions $N(x)$ and $G(x)$. First, taking the logarithm of $N(x)$, we obtain:

$$\ln N(x) = -\frac{T + 2 - i}{\tau} + \sum_{m=i-1}^{T} \ln \kappa_h(V[m], x) \qquad (39)$$

Taking the derivative of the above expression with respect to $x$, we obtain:

$$\frac{N'(x)}{N(x)} = \sum_{m=i-1}^{T} \frac{\partial \ln \kappa_h(V[m], x)}{\partial x} = \sum_{m=i-1}^{T} \frac{\kappa_h'(V[m], x)}{\kappa_h(V[m], x)} \qquad (40)$$

In the next step, we differentiate $G(x)$ and obtain:

$$G'(x) = \sum_{t=0}^{T}\sum_{i=0}^{t} e^{-\frac{i}{\tau}} \left(\prod_{j=1}^{i} \kappa_h(V[t-j], x) \cdot \sum_{k=1}^{i} \frac{\kappa_h'(V[t-k], x)}{\kappa_h(V[t-k], x)}\right) \qquad (41)$$

To compute $\frac{G'(x)}{G(x)}$, we can treat each term in $G(x)$ as the product of weights $w_{t,i}$ and then calculate the weighted average:

$$\frac{G'(x)}{G(x)} = \frac{\sum_{t=0}^{T}\sum_{i=0}^{t} w_{t,i} \cdot s_{t,i}}{G(x)} \qquad (42)$$

where

$$w_{t,i} = e^{-\frac{i}{\tau}} \prod_{j=1}^{i} \kappa_h(V[t-j], x) \qquad (43)$$

$$s_{t,i} = \sum_{k=1}^{i} \frac{\kappa_h'(V[t-k], x)}{\kappa_h(V[t-k], x)} \qquad (44)$$

Therefore,

$$\frac{G'(x)}{G(x)} = \frac{\sum_{t=0}^{T}\sum_{i=0}^{t} w_{t,i} \cdot s_{t,i}}{\sum_{t=0}^{T}\sum_{i=0}^{t} w_{t,i}} \qquad (45)$$

Next, we perform a piecewise analysis of $\frac{\kappa_h'}{\kappa_h}$ based on different intervals. First, according to Eq. (37), we have the following relationship:

$$\kappa'_h(V[m] - x) = \begin{cases} -V[m]\alpha^2, & -\frac{1}{\alpha} \leq V[m] - x \leq \frac{1}{\alpha} \\ 0, & \text{else} \end{cases} \tag{46}$$

Therefore, within the interval $-\frac{1}{\alpha} \leq V[m] - x \leq \frac{1}{\alpha}$, when $V[m] > 0$, $\frac{\kappa'_h(V[m],x)}{\kappa_h(V[m],x)} = \frac{-V[m]\alpha^2}{\kappa_h(V[m],x)} < 0$. Conversely, when $V[m] < 0$, $\frac{\kappa'_h(V[m],x)}{\kappa_h(V[m],x)} < 0$. Since the membrane potential follows a normal distribution with a mean of 0 (Guo et al., 2022b), when $x > 0$, we can assume that

$$\frac{N'(x)}{N(x)} = \sum_{m=i-1}^{T} \frac{\kappa'_h(V[m],x)}{\kappa_h(V[m],x)} < 0 \tag{47}$$

Moreover, due to the decreasing absolute values from $G_T$ to $G_1$, the logarithmic derivative of $N(x)$ is the sum of $\frac{\kappa'_h(V[m],x)}{\kappa_h(V[m],x)}$ from $i-1$ to $T$, while $G(x)$ represents the weighted average of $\frac{\kappa'_h(V[m],x)}{\kappa_h(V[m],x)}$ over the range from 1 to $T$. Therefore, we can consider that

$$\left| \frac{N'(x)}{N(x)} \right| > \left| \frac{G'(x)}{G(x)} \right| \tag{48}$$

Furthermore, since $N(x)$ is considered a subset of $G(x)$, they share the same sign from an expectation perspective. Therefore, we can assume that $F(x) = \frac{N(x)}{G(x)} > 0$. Combining Eqs. (47) and (48), we can draw the following conclusion:

$$\frac{N'(x)}{N(x)} - \frac{G'(x)}{G(x)} < 0 \tag{49}$$

Therefore,

$$F'(x) = F(x) \cdot \left( \frac{N'(x)}{N(x)} - \frac{G'(x)}{G(x)} \right) < 0 \tag{50}$$

This proves that $\Delta P_i - \Delta P_{i-1} < 0$ holds true, meaning that $\Delta \mathbf{P}$ is an increasing function. This demonstrates that LLIF neurons are capable of enhancing the contribution of gradients from earlier time steps to the overall gradient, thereby improving the neurons' ability to capture long-term dependencies.

## D ANALYSIS OF IMPACT OF $\tau$ IN SLIF'S GRADIENT

According to the gradient Eq. (51) for SLIF neurons, we can observe that the term affected by $\tau$ is $\exp\left(-\frac{i}{\tau}\right) \cdot \prod_{j=1}^{i} \kappa_s(V[t-j])$.

$$\nabla_{\mathbf{X}} \mathbf{S} = \sum_{t=0}^{T} \Theta'(V[t] - V_{\text{th}}) \cdot (1 + \sum_{i=0}^{t} \exp(-\frac{i}{\tau}) \cdot \prod_{j=1}^{i} \kappa_s(V[t-j])) \tag{51}$$

Based on this, in the case of a total of $T$ time steps, we can construct a function $G(\tau)$ with $T$ terms to analyze how $\tau$ influences the contribution of each term to the overall value of $G(\tau)$. The function $G(\tau)$ is defined as shown in Eq. (52), here, the notation of the terms in the equation follows the same rules as in Eq.(27).

$$G(\tau) = \underbrace{\exp(-\frac{1}{\tau})}_{G_T} + \underbrace{[\exp(-\frac{1}{\tau}) + \exp(-\frac{2}{\tau})]}_{G_{T-1}} + \cdots + \underbrace{[\exp(-\frac{1}{\tau}) + \exp(-\frac{2}{\tau}) + \cdots + \exp(-\frac{T}{\tau})]}_{G_1}$$

$$= \sum_{i=1}^{T}(T+1-i) \cdot \exp(-\frac{i}{x}) \tag{52}$$

Similar to Eq. 28, we construct a proportional distribution sequence $\mathbf{P}$, as shown in Eq. (53), for $G(\tau)$.

$$\mathbf{P} = [\frac{G_T}{G(\tau)}, \frac{G_{T-1}}{G(\tau)}, \cdots, \frac{G_1}{G(\tau)}] \tag{53}$$

When we adaptively adjust $\tau$ in the SLIF model using Eq. (9), and the resulting membrane time constant becomes $\tau'$, we can derive a new proportional distribution sequence $\mathbf{P}'$ based on $\tau'$. This sequence reflects the updated expected contribution of each time step to the total gradient.

$$\mathbf{P}' = [\frac{G'_T}{G(\tau')}, \frac{G'_{T-1}}{G(\tau')}, \cdots, \frac{G'_1}{G(\tau')}] \tag{54}$$

At this point, we compute the difference between $\mathbf{P}'$ and $\mathbf{P}$, resulting in the change sequence $\Delta\mathbf{P}$, as shown in Eq. (55). $\Delta\mathbf{P}$ represents the change in the contribution proportion of each time step to the gradient after the adaptive adjustment of $\tau$. Since both $\mathbf{P}$ and $\mathbf{P}'$ are based on the function $G(\tau)$, it is evident that they form increasing sequences, and all elements of both sequences are positive. Thus, to demonstrate that the gradient contribution proportion of recent time steps increases while that of early time steps decreases, we only need to prove that the $\Delta\mathbf{P}$ sequence is a decreasing sequence. This would confirm that adaptive $\tau$ strengthens the neuron's short-term dependency.

$$\Delta\mathbf{P} = \mathbf{P}' - \mathbf{P} = [\underbrace{\frac{G'_T}{G(\tau')} - \frac{G_T}{G(\tau)}}_{\Delta P_T}, \underbrace{\frac{G'_{T-1}}{G(\tau')} - \frac{G_{T-1}}{G(\tau)}}_{\Delta P_{T-1}}, \cdots, \underbrace{\frac{G'_1}{G(\tau')} - \frac{G_1}{G(\tau)}}_{\Delta P_1}] \tag{55}$$

In $\Delta\mathbf{P}$, following the labeling rule from $G(\tau)$ in Eq. (52), the first term $\frac{G'_T}{G(\tau')} - \frac{G_T}{G(\tau)}$ is labeled as $\Delta P_T$. To prove that $\Delta\mathbf{P}$ is a decreasing sequence, we need to show that for any $i \in [2, T]$, $\Delta P_i - \Delta P_{i-1} > 0$. Taking the difference between them gives:

$$\begin{aligned} \Delta P_i - \Delta P_{i-1} &= \frac{G'_i}{G(\tau')} - \frac{G_i}{G(\tau)} - (\frac{G'_{i-1}}{G(\tau')} - \frac{G_{i-1}}{G(\tau)}) \\ &= \frac{G'_i - G'_{i-1}}{G(\tau')} - \frac{G_i - G_{i-1}}{G(\tau)} \\ &= \frac{-\exp(-\frac{T+2-i}{\tau'})}{G(\tau')} - \frac{-\exp(-\frac{T+2-i}{\tau})}{G(\tau)} \end{aligned} \tag{56}$$

Based on the equation above, we construct the function as follows.

$$F(x) = \frac{-\exp(-\frac{T+2-i}{x})}{G(x)} \tag{57}$$

By taking the derivative of $F(x)$, we obtain:

$$
\begin{aligned}
\frac{dF(x)}{dx} &= \frac{\sum_{i=1}^{T}(T+1-i) \cdot \frac{i}{x^2} \cdot \exp(-\frac{T+2-i}{x}) \cdot \exp(-\frac{i}{x})}{(G(x))^2} \\
&\quad - \frac{\sum_{i=1}^{T}(T+1-i) \cdot \frac{T+2-i}{x^2} \cdot \exp(-\frac{T+2-i}{x}) \cdot \exp(-\frac{i}{x})}{(G(x))^2} \\
&= \frac{\exp(-\frac{T+2}{x})}{x^2 \cdot (G(x))^2} \cdot \sum_{i=1}^{T}(T+1-i) \cdot (2i-T-2) \\
&= \frac{\exp(-\frac{T+2}{x})}{x^2 \cdot (G(x))^2} \cdot (-\frac{T(T+1)(T+2)}{6}) < 0
\end{aligned}
\tag{58}
$$

Obviously, $F(x)$ is a monotonically decreasing function over the interval $x \in [0, +\infty)$. Thus, when applying adaptive membrane time constant, we have $\tau' < \tau$, which leads to $\Delta P_i - \Delta P_{i-1} > 0$. This shows that $\Delta \mathbf{P}$ forms a monotonically decreasing sequence, indicating that when adaptive $\tau$ is applied, the contribution of recent time steps to the gradient increases, while the contribution of early time steps decreases. Therefore, SLIF enhances the neuron's short-term dependency.

## E  DATASETS

**CIFAR-10.** The CIFAR-10 dataset (Krizhevsky et al., 2009) consists of 60,000 color images, each of size 32×32 pixels, divided into 10 different classes, such as airplanes, cars, birds, cats, and dogs. Each class has 6,000 images, with 50,000 images used for training and 10,000 for testing. Normalization, random horizontal flipping, random cropping with 4 padding, and CutOut (DeVries & Taylor, 2017) are applied for data augmentation.

**CIFAR-100.** The CIFAR-100 dataset (Krizhevsky et al., 2009) consists of 60,000 color images, each of size 32×32 pixels, categorized into 100 different classes. Each class contains 600 images, with 500 used for training and 100 for testing. The same processing methods as for dataset CIFAR-10 are applied to dataset CIFAR-100.

**Tiny-ImageNet.** The Tiny-ImageNet dataset is a scaled-down version of the ImageNet dataset (Deng et al., 2009). It contains 200 different classes, with 500 training images and 50 testing images per class, resulting in a total of 100,000 training images and 10,000 testing images. Each image is resized to 64×64 pixels. Normalization, random horizontal flipping, and random cropping with 4 padding are applied for data augmentation for the Tiny-ImageNet dataset.

**DVS-CIFAR-10.** The DVS-CIFAR-10 dataset (Li et al., 2017) is a neuromorphic version of the traditional CIFAR-10 dataset. DVS-CIFAR-10 captures the visual information using a Dynamic Vision Sensor (DVS), which records changes in the scene as a series of asynchronous events rather than as a sequence of frames. The dataset consists of recordings of 10 object classes, corresponding to the original CIFAR-10 categories, with each object presented in front of a DVS camera under various conditions. The dataset contains 10,000 128×128 images, of which 9,000 are used as the training set and the remaining 1,000 as the test set.

**DVS-Gesture.** The DVS-Gesture dataset (Amir et al., 2017) is a neuromorphic dataset, consisting of 11 different hand gesture classes, such as hand clapping, arm rolling, and air guitar, performed by 29 subjects under various lighting conditions. Each gesture is represented by a sequence of events rather than frames. The dataset contains 1,176 training samples and 288 testing samples.

## F  SEARCHING AND RETRAINING SETTINGS

The code for this work is implemented using PyTorch (Paszke et al., 2019) and SpikingJelly framework (Fang et al., 2023).

The ST-DANO process is divided into two phases: searching and retraining. All experiments are conducted using a NVIDIA RTX 3090 GPU. The hyperparameters used in each phase are detailed in Table 4 and Table 5, respectively. Specifically, in the searching phase, for all datasets, half of the

training set is used as training data (to update weights), while the other half is used as validation data (to update architecture parameters). We ensure that the retraining hyperparameters involved during searching phase are set exactly the same as those used in the actual retraining phase.

Table 4: Hyperparameter settings for searching phase. a-lr and a-wd denote the learning rate and weight decay factor for architecture parameters, respectively, and bs is the batch size.

| Dataset | optimizer | a-lr | a-wd | epoch | bs |
|---------|-----------|------|------|-------|-----|
| CIFAR-10 | Adam | 3e-4 | 1e-4 | 100 | 128 |
| CIFAR-100 | Adam | 3e-4 | 1e-4 | 100 | 128 |
| Tiny-ImageNet | Adam | 3e-4 | 1e-4 | 100 | 128 |
| DVS-CIFAR-10 | Adam | 1e-3 | 5e-4 | 100 | 128 |
| DVS-Gesture | Adam | 1e-3 | 5e-4 | 100 | 8 |

Table 5: Hyperparameter settings for retraining phase. lr and wd denote the learning rate and weight decay factor, respectively, bs is the batch size, and dr is the dropout rate.

| Dataset | optimizer | lr | wd | epoch | bs | dr |
|---------|-----------|-----|------|-------|-----|-----|
| CIFAR-10 | SGD | 0.1 | 5e-5 | 300 | 128 | 0 |
| CIFAR-100 | SGD | 0.1 | 5e-4 | 300 | 128 | 0 |
| Tiny-ImageNet | SGD | 0.1 | 5e-4 | 300 | 128 | 0 |
| DVS-CIFAR-10 | SGD | 0.05 | 5e-4 | 300 | 128 | 0.3 |
| DVS-Gesture | SGD | 0.05 | 5e-4 | 300 | 8 | 0.4 |

Additionally, we adopt temporal efficient training (TET) loss function (Deng et al., 2022) for re-training phase:

$$\mathcal{L} = \frac{1}{T}\sum_{t=1}^{T}(1-\lambda)\mathcal{L}_{CE}(\mathbf{O}[t], \mathbf{y}) + \lambda\mathcal{L}_{MSE}(\mathbf{O}[t], V_{\text{th}}) \tag{59}$$

where $\mathcal{L}_{CE}$ and $\mathcal{L}_{MSE}$ refer to the cross-entropy function and mean squared error function, respectively. The proportion of the regular term is controlled by a hyperparameter $\lambda$, which is set to 0.05 for all cases in our experiments.

## G  EXTENSION OF ABLATION STUDY

This section provides the full version of the curves from the ablation experiments presented in Section 4.2. Due to the large number of methods involved in the ablation experiments, the overlapping of curves made visualization challenging. To address this, we present each curve in a separate plot with bold colors for clarity, while lighter-colored curves represent other methods within the same plot for comparison. The ablation experiments on the CIFAR-10 dataset are shown in Fig. 4, and those on the DVS-CIFAR-10 dataset are displayed in Fig. 5. In the ablation experiments on CIFAR-10, we set the number of epochs to 500 instead of 300 to better illustrate the differences between the methods. As for the other hyperparameter settings in this ablation experiments, we adhered to the configurations used in the retraining phase, which are given in Appendix F.

## H  VISUALIZATION FOR OPTIMAL NEURON CONFIGURATIONS

In this section, we present the visualization (Heatmaps of architecture parameters) for optimal neuron configurations found by ST-DANO (Figs. 6, 7, 8, and 9). As can be seen, for the same dataset, the optimal neuron configurations identified by ST-DANO under different time step settings exhibit only minor differences in a few network layers, while overall they remain quite similar. This indicates that the ST-DANO search process is relatively stable and effectively explores the search space.

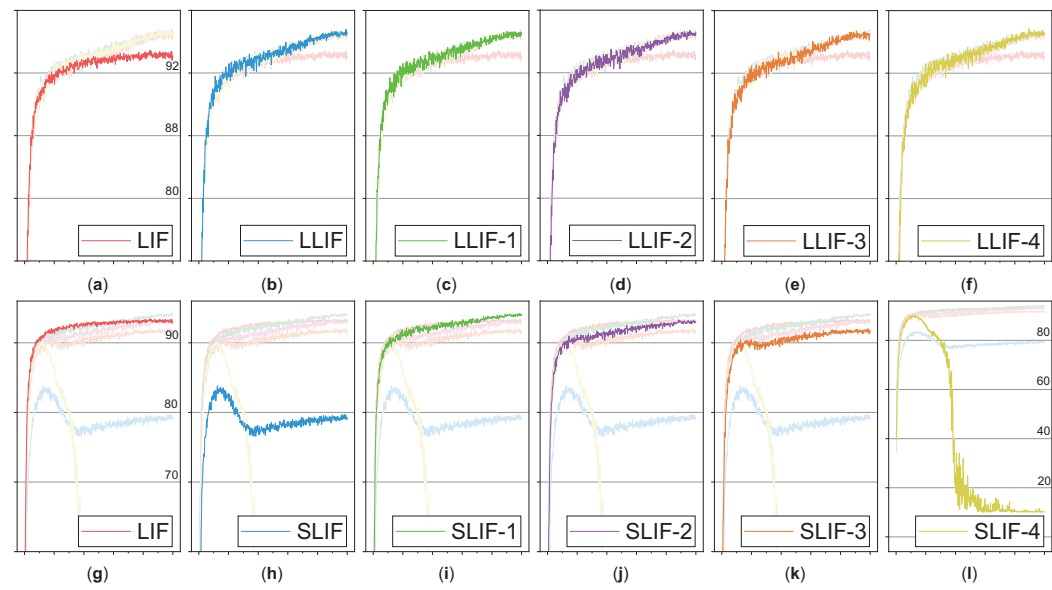

Figure 4: Test accuracy curves of ablation study on CIFAR-10. Where the X-axis of all plots is epoch, identically ranging from 0 to 500, and the Y-axis of all plots is the test accuracy (%), where the (b)-(f) plots have the same range in Y-axis as (a), and the (h)-(k) plots go with the same range in Y-axis as (g).

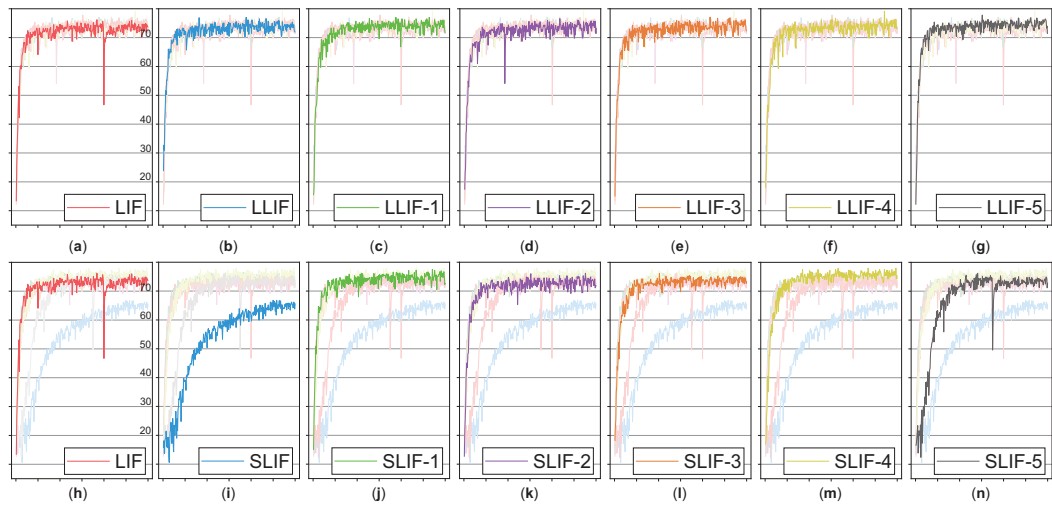

Figure 5: Test accuracy curves of ablation study on DVS-CIFAR-10. Where the X-axis of all plots is epoch, identically ranging from 0 to 300, and the Y-axis of all plots is the test accuracy (%), where the (b)-(g) plots have the same range in Y-axis as (a), and the (i)-(n) plots go with the same range in Y-axis as (h).

## I FIRE RATE EVALUATION

Although both LLIF and SLIF introduce additional computations compared to LIF during inference, these computations consist only of a small number of extra additions and $T$ constant-matrix multiplications during the forward propagation process ($T$ is the number of time steps). The extra energy burden from these operations is nearly negligible. Furthermore, from the perspective of spike firing rates, which are given in Fig. 10, the architectures identified by ST-DANO exhibit lower average

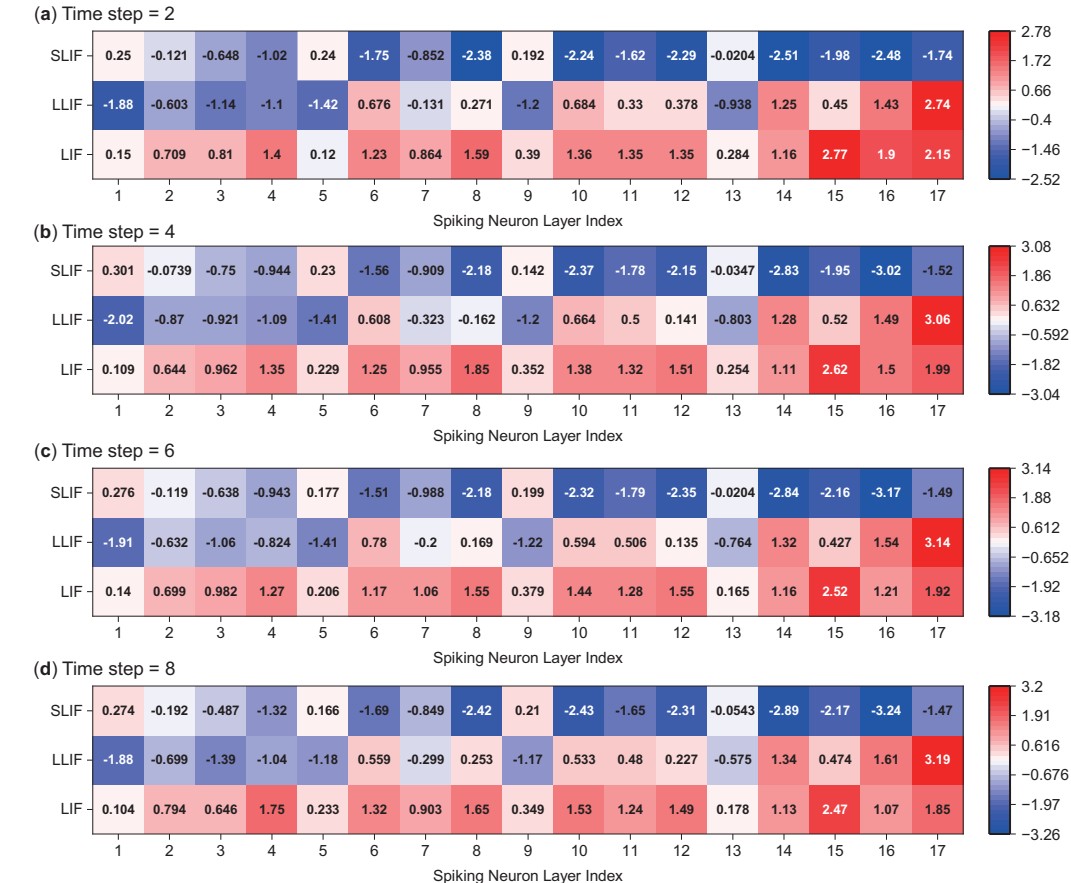

Figure 6: Architecture parameters visualization across multiple time step settings on CIFAR-10 (ResNet-18).

firing rates across all datasets compared to those based on LIF. This is because both LLIF and SLIF neurons are designed to suppress excessive neuron excitation and limit spike firings. In summary, combining the above analysis and experiments, ST-DANO-optimized architectures are able to improve algorithmic performance while conserving energy consumption.

## J  ADDITIONAL EXPERIMENTS

In this section, we present the performance of ST-DANO across various base network architectures, comparing these results with other methods using the same architectures (Appendix J.1). Additionally, we provide details on the experimental setup used for these additional experiments (Appendix J.2), along with the optimal frameworks identified through our search (Appendix J.3).

### J.1  PERFORMANCE OF EXTENDED CASES

In Table 6, we present the experimental results of ST-DANO obtained after searching across different base network architectures. Compared to the experiments in the main body, the additional experiments include: VGG-16 on CIFAR-10, VGG-16 on CIFAR-100, ResNet-18 on Tiny-ImageNet, ResNet-18 on DVS-CIFAR-10, and ResNet-18 on DVS-Gesture.

From these additional experiments, we can conclude that ST-DANO consistently achieves competitive performance across various base architectures, demonstrating its generalizability.

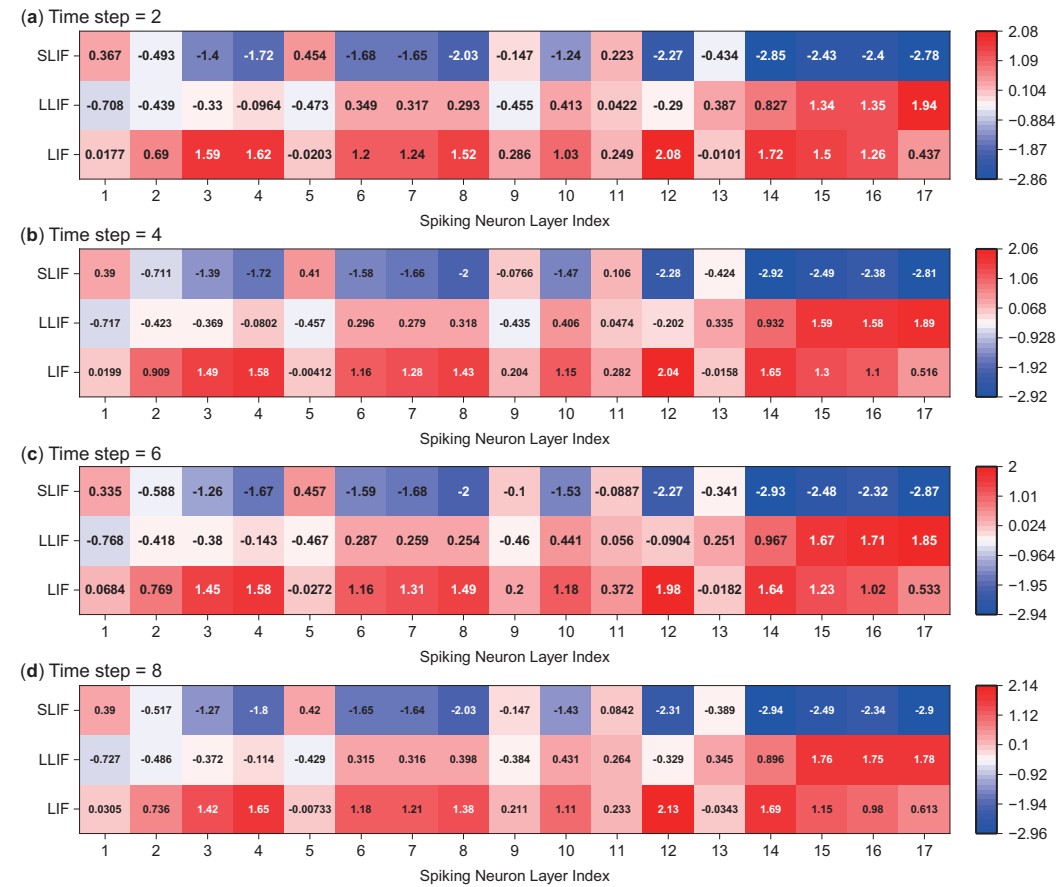

Figure 7: Architecture parameters visualization across multiple time step settings on CIFAR-100 (ResNet-18).

## J.2 Searching and Retraining settings for Additional Experiments

We conduct the additional experiments using the same hardware configuration as in the main experiments. Due to GPU memory limitations, we proportionally reduced the batch size and learning rate for some cases during the search phase of the additional experiments. The hyperparameters for the search phase in additional experiments are detailed in Table 7, while the retraining phase followed the exact hyperparameters listed in Table 5.

## J.3 Optimal Neuron Configurations of Extended Cases

This section presents the optimal neuron configurations obtained by ST-DANO across various datasets and under different time step settings in the additional experiments (Table 8).

## K Reproducibility Statement

The complete code with fixed random seed utilized in this work is provided in the supplementary materials and will be made publicly available after this paper is published. All datasets employed in this research, including CIFAR-10, CIFAR-100, Tiny-ImageNet, DVS-CIFAR-10, and DVS-Gesture, are publicly accessible. Details regarding the hardware, coding environment, and hyperparameter settings used in our experiments are also included in the Appendix. We dedicate to enable future researchers to replicate the results presented in this paper using similar computational setups.

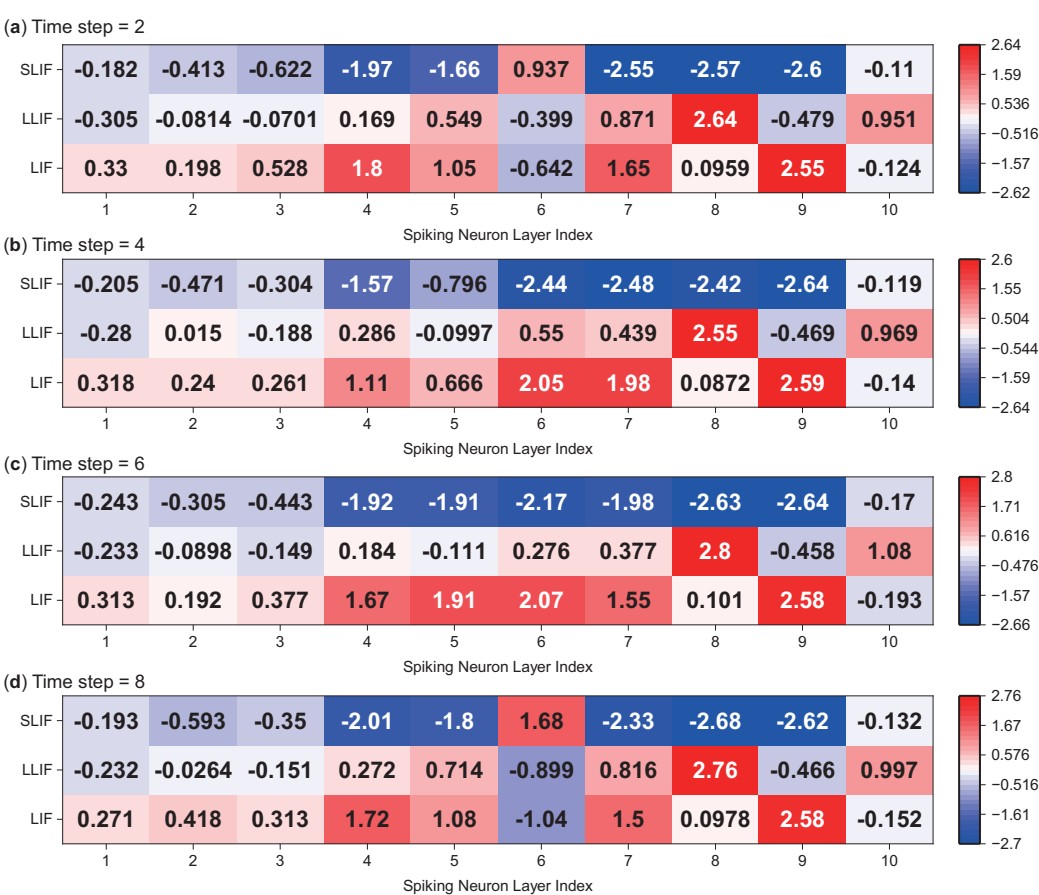

Figure 8: Architecture parameters visualization across multiple time step settings on Tiny-ImageNet (VGG-13).

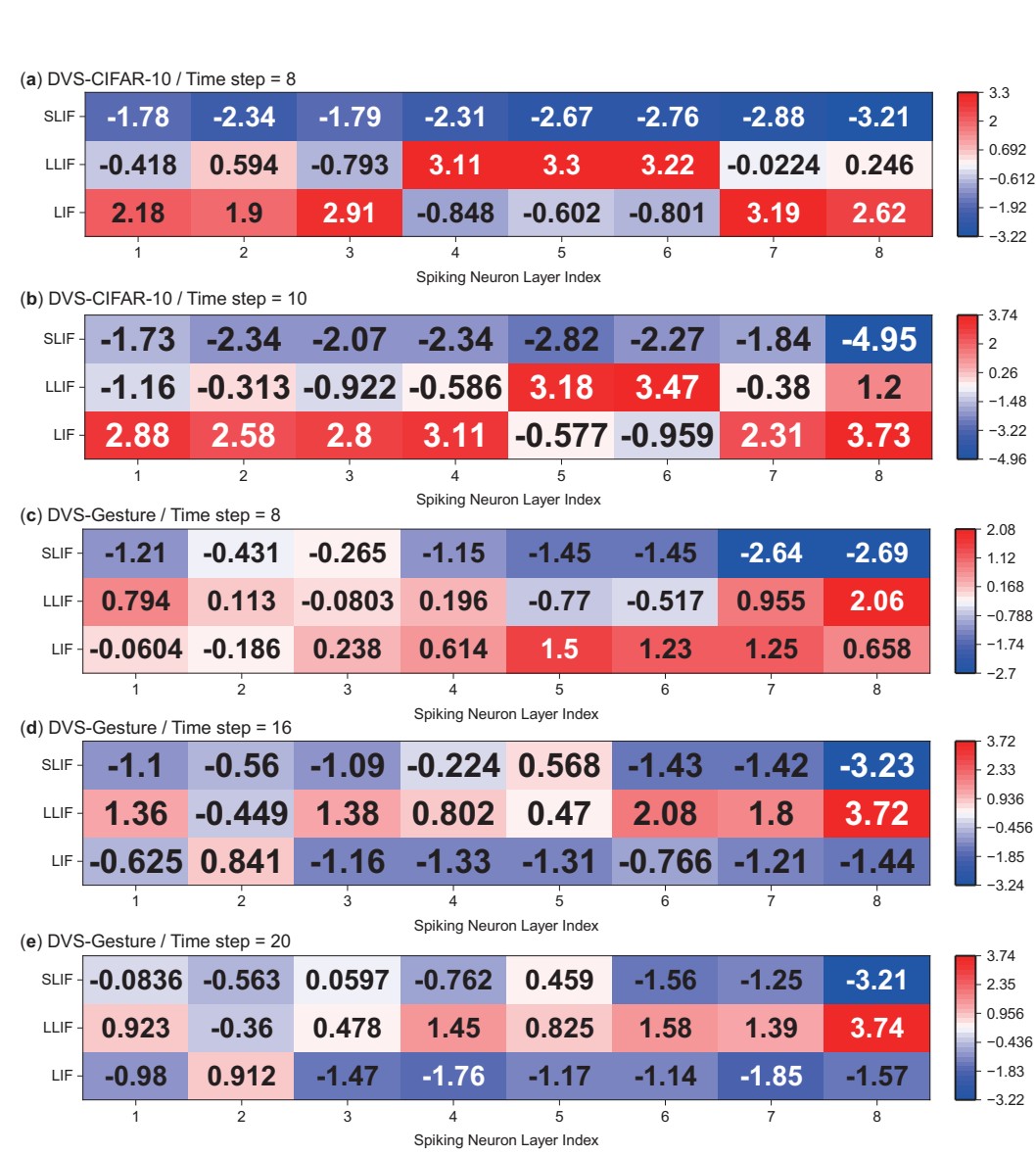

Figure 9: Architecture parameters visualization across multiple time step settings on DVS-CIFAR-10 and DVS-Gesture (VGG-11).

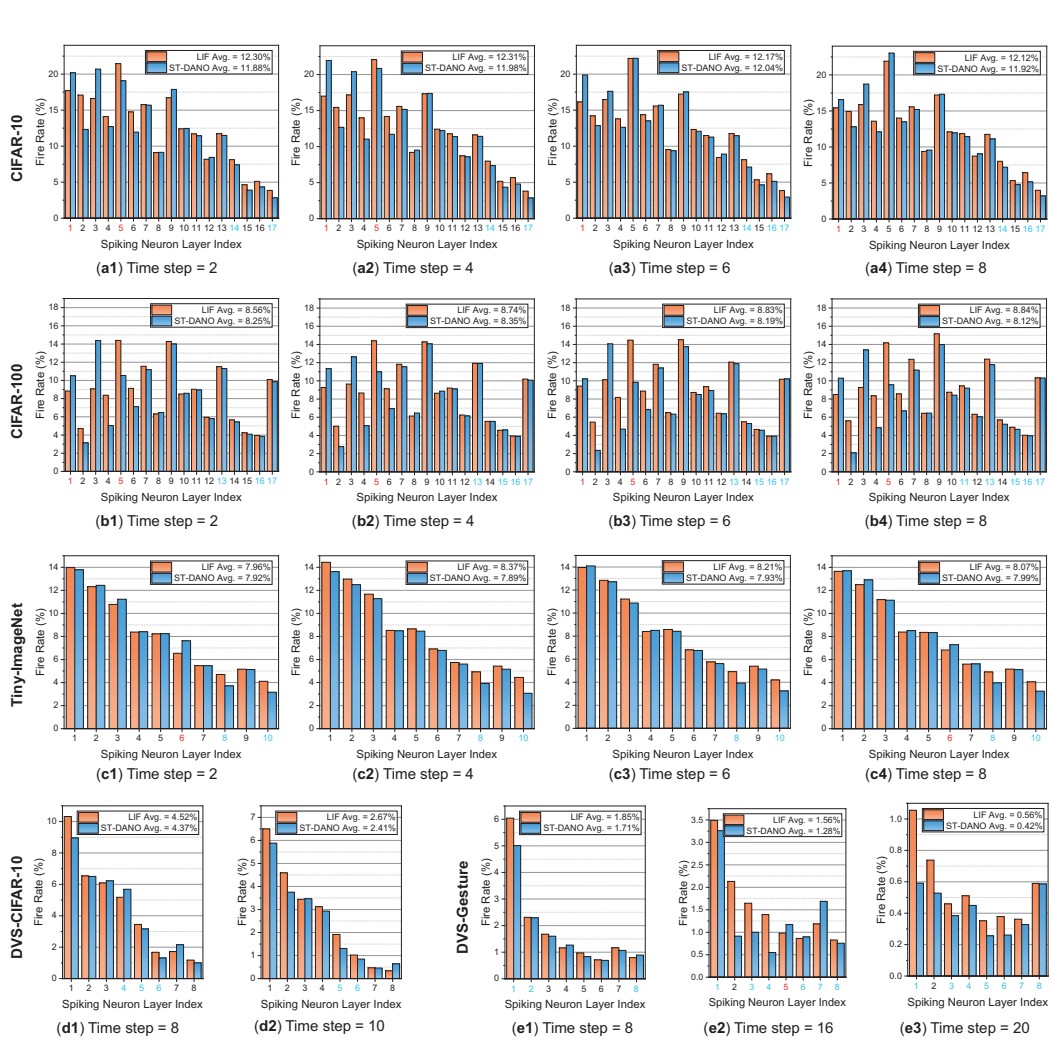

Figure 10: Fire rate evaluation. The black numbers on the x-axis indicate that the ST-DANO architecture uses LIF in that network layer, blue denotes LLIF, and red represents SLIF, consistent with the optimal neuron configurations shown in Table 3.

Table 6: Comparison using the same architecture with SOTA on CIFAR-10, CIFAR-100, Tiny-ImageNet, DVS-CIFAR-10, and DVS-Gesture datasets.

| | Model | Architecture | Timestep | Accuracy (%) |
|---|---|---|---|---|
| CIFAR-10 | Joint A-SNN (Guo et al., 2023a) | ResNet-18 | 4 | 95.45 |
| | TET-SSF (Wang et al., 2023a) | | 20 | 94.90 |
| | SLTT (Meng et al., 2023) | | 6 | 94.59 |
| | **ST-DANO (Ours)** | | 8 / 6 / 4 / 2 | **95.92 / 95.65 / 95.66 / 95.50** |
| | Te. Pr. (Chowdhury et al., 2022) | VGG-16 | 5 | **93.90** |
| | DIET (Rathi & Roy, 2021) | | 10 | 93.44 |
| | Offline LTL (Yang et al., 2022) | | 16 | 93.23 |
| | **ST-DANO (Ours)** | | 8 / 6 / 4 / 2 | 93.87 / 93.06 / 92.65 / 91.98 |
| CIFAR-100 | Joint A-SNN (Guo et al., 2023a) | ResNet-18 | 4 | **77.39** |
| | RecDis-SSF (Wang et al., 2023a) | | 20 | 75.48 |
| | sparse-KD (Xu et al., 2024b) | | 4 | 73.01 |
| | SLTT (Meng et al., 2023) | | 6 | 74.67 |
| | **ST-DANO (Ours)** | | 8 / 6 / 4 / 2 | 76.25 / 75.94 / 75.93 / 75.53 |
| | Te. Pr. (Chowdhury et al., 2022) | VGG-16 | 5 | 71.58 |
| | DIET (Rathi & Roy, 2021) | | 10 | 69.67 |
| | Offline LTL (Yang et al., 2022) | | 16 | 74.19 |
| | **ST-DANO (Ours)** | | 8 / 6 / 4 / 2 | **74.33 / 74.21** / 73.78 / 72.99 |
| Tiny-ImageNet | default-KD (Xu et al., 2024b) | ResNet-18 | 4 | 59.31 |
| | STC (Xu et al., 2024a) | | 8 | 59.99 |
| | **ST-DANO (Ours)** | | 8 / 6 / 4 / 2 | **60.21 / 60.16** / 59.68 / 59.23 |
| | DCT (Garg et al., 2020) | VGG-13 | 125 | 56.90 |
| | ASGL (Wang et al., 2023b) | | 8 | 56.81 |
| | Offline LTL (Yang et al., 2022) | | 32 | 55.85 |
| | **ST-DANO (Ours)** | | 8 / 6 / 4 / 2 | **60.24 / 60.14 / 60.08 / 59.10** |
| DVS-CIFAR-10 | SLTT (Meng et al., 2023) | ResNet-18 | 10 | 77.30 |
| | InfLoR (Guo et al., 2022a) | | 10 | 75.50 |
| | DiffSpike (Li et al., 2021b) | | 10 | 75.40 |
| | **ST-DANO / ST-DANO† (Ours)** | | 10 | **76.90 / 79.90** |
| | CE-SSF (Wang et al., 2023a) | VGG-11 | 20 | 78.00 |
| | OTTT (Xiao et al., 2022) | | 10 | 77.10 |
| | **ST-DANO / ST-DANO† (Ours)** | | 10 | **79.30 / 83.90** |
| | **ST-DANO / ST-DANO† (Ours)** | | 8 | 77.20 / 81.80 |
| DVS-Gesture | tdBN (Zheng et al., 2021) | ResNet-18 | 40 | 96.87 |
| | SLTT (Meng et al., 2023) | | 20 | **98.26** |
| | **ST-DANO (Ours)** | | 20 | 97.22 |
| | OTTT (Xiao et al., 2022) | VGG-11 | 20 | 96.88 |
| | SLTT (Meng et al., 2023) | | 20 | **98.62** |
| | **ST-DANO (Ours)** | | 20 / 16 / 8 | 98.61 / 97.22 / 96.88 |

† Data augmentation (Li et al., 2022).

Table 7: Hyperparameter settings for searching phase in additional experiments. a-lr and a-wd denote the learning rate and weight decay factor for architecture parameters, respectively, and bs is the batch size.

| Dataset | optimizer | a-lr | a-wd | epoch | bs |
|---|---|---|---|---|---|
| CIFAR-10 | Adam | 3e-4 | 1e-4 | 100 | 128 |
| CIFAR-100 | Adam | 3e-4 | 1e-4 | 100 | 128 |
| Tiny-ImageNet | Adam | 1.5e-4 | 1e-4 | 100 | 64 |
| DVS-CIFAR-10 | Adam | 2.5e-4 | 5e-4 | 100 | 32 |
| DVS-Gesture | Adam | 5e-4 | 5e-4 | 100 | 4 |

Table 8: Optimal neuron configurations discovered by ST-DANO across different datasets in additional experiments. The sequence of numbers in the 'Optimal Neuron Configuration' represents the types of neurons used from the input layer to the output layer of the network, where 0 denotes LIF neurons, 1 denotes LLIF neurons, and 2 denotes SLIF neurons.'Search Cost' is measured in GPU days.

| Dataset | Architecture | Time Step | Optimal Neuron Configuration | Search Cost |
|---|---|---|---|---|
| CIFAR-10 | VGG-16 | 2 | 2002000111110 | 0.09 |
| | | 4 | 2000000000110 | 0.11 |
| | | 6 | 2201000001110 | 0.13 |
| | | 8 | 2002000011110 | 0.15 |
| CIFAR-100 | VGG-16 | 2 | 2011000000000 | 0.09 |
| | | 4 | 0011000000020 | 0.11 |
| | | 6 | 0011000000020 | 0.13 |
| | | 8 | 0011000000000 | 0.15 |
| Tiny-ImageNet | ResNet-18 | 2 | 22002001100112111 | 0.74 |
| | | 4 | 22002000100100111 | 0.91 |
| | | 6 | 22002001100110011 | 1.20 |
| | | 8 | 22202000100100001 | 1.43 |
| DVS-CIFAR-10 | ResNet-18 | 10 | 10121010002101111 | 0.74 |
| DVS-Gesture | ResNet-18 | 20 | 11011100111001111 | 0.36 |

