# OpenReview forum: "Spatio-Temporal Dependency-Aware Neuron Optimization for Spiking Neural Networks"
_ICLR.cc/2025/Conference — Submitted to ICLR 2025_

### Official Review · Reviewer_B74X · 2024-10-22

**Soundness:** 3
**Presentation:** 3
**Contribution:** 2
**Rating:** 5
**Confidence:** 5

**Summary:**

This paper proposes two variants of the LIF neuron, Long-LIF and Short-LIF. An ablation study is carried out to check neurons' prefered depth in the SNN. Then a NAS-based method is used to search the neuron type. The proposed methods are validated on various datasets.

**Strengths:**

This paper is written clearly and easy to be understood.
The experiments and details are abundant.

**Weaknesses:**

The proposed neurons use tanh activation in neuronal dynamics (Eq.7 and Eq.9), which involves exponential operations and can hardly be implemented in neuromorphic hardware. While previous methods often use different train/inference behaviors to avoid this problem. For example, the attention SNN uses sigmoid in training and Heaviside in inference. (Please refer to `Temporal-wise Attention Spiking Neural Networks for Event Streams Classification`). I suggest that the authors consider similar solutions.


In lines 232-247, the authors try to explain how the increasing of v_th will influence the gradient. This explaination is very weak and ambiguous. It is better to use a quantitative analysis. For example, the authors could show the derivative of Eq.4 with respect to V_th to prove the monotonicity of the gradient.

The design of Eq.6 and Eq.7 for the Long-LIF is discussed clearly that the negative inputs are regarded as the inhibitory signals and will increase the threshold. However, the motivation of Eq.9 for the Short-LIF neuron is not introduced. According to Eq.8, a large V_p will cause a smaller tau. How the short-term dependency is reflected in the neuronal dynamics?


The proposed neuron models focus on long/short-term dependency. In this context, the results of the experiment on memory tasks are more desirable. Previous research  such as `Accurate and efficient time-domain classification with adaptive spiking recurrent neural networks` uses the sequential/pixel classification to evaluate the learning ability of proposed models. I suggest that the authors report these results with comparison to previous neuron models.

**Questions:**

Please refer to `Weaknesses`.

---

> ### Author Response · Authors · 2024-11-19
> **Response to Reviewer B74X (Part 1/4)**
>
> First and foremost, we sincerely appreciate your positive evaluation of the strengths of our work. Below, we have provided responses to the concerns you raised.
>
> **Weakness 1: The proposed neurons use tanh activation in neuronal dynamics (Eq.7 and Eq.9), which involves exponential operations and can hardly be implemented in neuromorphic hardware. While previous methods often use different train/inference behaviors to avoid this problem. For example, the attention SNN uses sigmoid in training and Heaviside in inference. (Please refer to Temporal-wise Attention Spiking Neural Networks for Event Streams Classification). I suggest that the authors consider similar solutions.**
>
> We understand your concerns. Our next step is to deploy the model onto neuromorphic hardware, and therefore, we conducted extensive research in this area prior to designing our approach. Below, we will elaborate on several aspects that demonstrate the feasibility of deploying our method on neuromorphic hardware without any loss in performance.
>
> 1. **Exponential operations over a limited domain can be implemented on neuromorphic hardware using a lookup table approach.** In our method, both the membrane potential threshold and the membrane time constant are adaptively adjusted using Eqs. 7 and 9, respectively, which involve the exponential function tanh. In Eq. 7, the variable for tanh can be approximately interpreted as the input current to the neuron, with values distributed in the range of [-2, 2] and following a Gaussian distribution [1]. For Eq. 9, the variable of tanh represents the residual membrane potential $V_p$, where $V_p$ lies within [0, 1]. On the other hand, considering the definition of tanh, it implies that to efficiently compute tanh using a table-based approach, we only need to store discrete values of $e^x$ and $e^{-x}$ over the range of [-2, 2]. This allows for fast computation of tanh without involving complex exponential operations. This leads to a related question: what is the maximum level of precision we can achieve when discretizing the interval [-2, 2]? We will provide a detailed analysis based on existing neuromorphic hardware capabilities in the following sections.
>
> 2. **Neuromorphic hardware available on the market can accommodate the memory overhead required for the lookup table approach.** For example, Intel's neuromorphic chip Loihi 2, introduced in 2021, features 192KB of flexibly allocatable memory, as described on page 7 of the product brief available at [2] ([Link](https://download.intel.com/newsroom/2021/new-technologies/neuromorphic-computing-loihi-2-brief.pdf)). With 192KB of memory, it is possible to store up to 24,576 double-precision floating-point numbers. This allows for discretizing the range [-2, 2] for $e^x$ and $e^{-x}$ with a maximum precision of $3.33×10^{-4}$, enabling their respective storage. Next, we will further analyze how this level of precision impacts the performance of the algorithm.
>
> 3. **Result: the loss in computational precision introduced by the lookup table approach has no impact on the performance of our method.** To investigate how reducing the precision of the input variables for the tanh function in Eqs. 7 and 9 to three or four decimal places affects algorithm performance, we conducted experiments during the inference phase of the model. Specifically, we rounded the inputs of the tanh function to explore the performance impact across different precision levels (with varying numbers of decimal places retained). The tables below show the changes in algorithm performance. "Text" denotes the case where the tanh input precision is unmodified (identical to the precision reported in the main text), while 1. to 4. represent scenarios where the tanh precision is controlled at 1 to 4 decimal places, respectively. Any values different from "Text" are highlighted.

---

> ### Author Response · Authors · 2024-11-19
> **Response to Reviewer B74X (Part 2/4)**
>
> **(Continue to response Weakness 1)**
>
>
> | Dataset        | Timestep  | text   | 1.       | 2.       | 3.       | 4.       |
> |----------------|----|--------|---------|---------|---------|---------|
> | Cifar 10       | 2  | 95.50  | **95.48** | 95.50   | 95.50   | 95.50   |
> |                | 4  | 95.66  | 95.66   | 95.66   | 95.66   | 95.66   |
> |                | 6  | 95.65  | **95.66** | 95.65   | 95.65   | 95.65   |
> |                | 8  | 95.92  | 95.92   | 95.92   | 95.92   | 95.92   |
> | | | | | | | |
> | Cifar 100      | 2  | 75.18  | **75.05** | 75.18   | 75.18   | 75.18   |
> |                | 4  | 75.18  | **75.93** | 75.18   | 75.18   | 75.18   |
> |                | 6  | 75.94  | **75.98** | 75.94   | 75.94   | 75.94   |
> |                | 8  | 76.25  | **76.15** | **76.21** | 76.25 | 76.25   |
> | | | | | | | |
> | Tiny-ImageNet  | 2  | 59.10  | **59.22** | **59.00** | 59.10 | 59.10   |
> |                | 4  | 60.08  | **59.96** | **60.06** | 60.08 | 60.08   |
> |                | 6  | 60.14  | **60.16** | 60.14   | 60.14   | 60.14   |
> |                | 8  | 60.24  | **60.26** | 60.24   | 60.24   | 60.24   |
> | | | | | | | |
> | DVS-CIFAR-10   | 8  | 81.80  | **81.90** | **81.90** | 81.80 | 81.80   |
> |                | 10 | 83.90  | **83.60** | 83.90   | 83.90   | 83.90   |
> | | | | | | | |
> | DVS-Gesture    | 8  | 96.88  | **96.53** | **96.53** | 96.88 | 96.88   |
> |                | 16 | 97.22  | **97.57** | **97.57** | 97.22 | 97.22   |
> |                | 20 | 98.61  | 98.61   | 98.61   | 98.61   | 98.61   |
>
> From the above experiments, we observe that the algorithm's performance is only minimally affected when the tanh input retains one or two decimal places, and this impact is not necessarily detrimental to precision. In some cases, such as CIFAR-100 (T=4) with a 0.75% increase, Tiny-ImageNet (T=2) with a 0.12% increase, and DVS-Gesture (T=16) with a 0.35% increase, the model's performance even surpasses the baseline when retaining one decimal place. For cases where three or four decimal places are retained, the model consistently maintained its performance across all datasets and time step configurations. This demonstrates that, using the Loihi 2 chip as an example, our model can utilize the lookup table approach for computing tanh within the constrained memory size, enabling lossless deployment of the model. Actually, this phenomenon is understandable given the characteristics of SNNs. The forward propagation process in SNNs involves multiple conversions between continuous and discrete values, leading to unavoidable cumulative errors during forward propagation. By using different decimal precision levels for tanh, we introduce only a slight increase in cumulative error, which is dwarfed by the substantial inherent error accumulation in SNN forward propagation. Therefore, this behavior has an extremely minimal impact on the overall model performance. Next, we present existing works that utilize neuromorphic hardware to implement complex nonlinear operations, further supporting our conclusion.
>
> 4. **Existing cases of complex nonlinear function computation using lookup tables.** For example, the 5-th page in [3]:
>
> > "Alternatively, Tianjic [4] (a hybrid neuromorphic chip) discretized and iterated the ODE solving process with time steps and adopted a lookup table to realize nonlinear function solving, leading to the improvement in performance."
>
> Other examples include [5, 6].
>
> 5. **Summary.** Based on the above analysis, we conclude that, considering the products currently available on the market, **the tanh computation in our method can be implemented on neuromorphic hardware using a lookup table approach**. Furthermore, taking the Loihi2 chip as an example, the discrete lookup table method introduces **no accuracy loss in model deployment** observed in our experiments above.
>
>
> > [1] Guo, Yufei, et al. "Recdis-snn: Rectifying membrane potential distribution for directly training spiking neural networks." CVPR, 2022.
>
> > [2] https://download.intel.com/newsroom/2021/new-technologies/neuromorphic-computing-loihi-2-brief.pdf
>
> > [3] Yu, Fangwen, et al. "Brain-inspired multimodal hybrid neural network for robot place recognition." Science Robotics, 2023.
>
> > [4] Pei, Jing, et al. "Towards artificial general intelligence with hybrid Tianjic chip architecture." Nature, 2019.
>
> > [5] Siddique, Ali, Mang I. Vai, and Sio Hang Pun. "A low cost neuromorphic learning engine based on a high performance supervised SNN learning algorithm." Science Robotics, 2023.
>
> > [6] Siddique, Ali, Mang I. Vai, and Sio Hang Pun. "A low-cost, high-throughput neuromorphic computer for online SNN learning." Cluster Computing, 2024.

---

> ### Author Response · Authors · 2024-11-19
> **Response to Reviewer B74X (Part 3/4)**
>
> **Weakness 2: In lines 232-247, the authors try to explain how the increasing of v_th will influence the gradient. This explaination is very weak and ambiguous. It is better to use a quantitative analysis. For example, the authors could show the derivative of Eq.4 with respect to V_th to prove the monotonicity of the gradient.**
>
> Thank you very much for pointing out this issue, as it is highly beneficial for improving the quality of our work. In our previous manuscript, we adopted a qualitative analysis, and we acknowledge that this approach was not sufficiently rigorous. Therefore, in the revised version, we have incorporated your suggested approach by deriving Eq. 4 with respect to $V_{th}$ and proving its monotonicity. The conclusion indicates that LLIF neurons can enhance long-term dependencies on input data. However, due to the length of the derivation, we apologize for not including it here; please refer to Appendix C, lines 882-1065, which is an added section with highlighted in blue, of the revised manuscript for the detailed analysis.
>
> **Weakness 3: The design of Eq.6 and Eq.7 for the Long-LIF is discussed clearly that the negative inputs are regarded as the inhibitory signals and will increase the threshold. However, the motivation of Eq.9 for the Short-LIF neuron is not introduced. According to Eq.8, a large V_p will cause a smaller tau. How the short-term dependency is reflected in the neuronal dynamics?**
>
> Thank you for your valuable feedback. In fact, our design in SLIF contrasts with that of the LLIF model. In LLIF, we considered negative membrane potential as inhibitory signals and thus adapted the threshold accordingly. In SLIF, however, we regard overly high positive potentials as over-activation signals. Such potentials lead to higher residual membrane potentials under a soft reset mechanism, where the magnitude of the residual membrane potential reflects the strength of recent neuronal excitation. Furthermore, based on this excitation strength, we leverage the phenomenon of synaptic potentiation observed in biological neurons [1]—where neurons temporarily increase sensitivity to recent signals under strong stimulation while moderately reducing the influence of historical activations—and correlate this with the adaptation of the membrane time constant. This mechanism resembles a form of short-term memory effect.
>
> Specifically, we have enhanced the description of this motivation as follows:
>
> > **Short-LIF (SLIF).** Strengthening short-term dependency requires neurons to enhance their sensitivity to recent time steps. Based on the synaptic potentiation phenomenon (Zucker & Regehr, 2002) observed in biological neurons—where a neuron temporarily increases its sensitivity to recent signals when subjected to strong stimulation while moderately reducing the influence of historical activations—the design in SLIF neurons contrasts with the inhibitory signal design in LLIF. In SLIF, we treat membrane potentials exceeding the threshold as signals of over-excitation, linking this signal to the sensitivity of signal reception in future time steps. This implies that in SLIF neurons, the rate of decay for past accumulated potentials at the current time step is positively correlated with the intensity of recent stimulation, enabling the neuron to pay more attention to recent inputs. Specifically, in SLIF, we employ a soft reset method (detailed in Appendix A, where the degree of recent stimulation is represented by the residual membrane potential, which refers to the neuron's potential after a soft reset. The decay rate of the potential is controlled by $\tau$. Thus, the SLIF process, as depicted in Fig. (1)(b), defines the residual membrane potential $V_p$ as per Eq. (8), while the dynamic relationship between the membrane time constant $\tau$ and the residual membrane potential is given by Eq. (9).
>
> In the revised manuscript, this section has been highlighted in blue, located on lines 233-245.
>
> > [1] Zucker, Robert S., and Wade G. Regehr. "Short-term synaptic plasticity." Annual review of physiology, 2002.

---

> ### Author Response · Authors · 2024-11-19
> **Response to Reviewer B74X (Part 4/4)**
>
> **Weakness 4: The proposed neuron models focus on long/short-term dependency. In this context, the results of the experiment on memory tasks are more desirable. Previous research such as Accurate and efficient time-domain classification with adaptive spiking recurrent neural networks uses the sequential/pixel classification to evaluate the learning ability of proposed models. I suggest that the authors report these results with comparison to previous neuron models.**
>
> Thank you for this suggestion. Our method is based on the BPTT + direct encoding approach, which we have explicitly stated in the Introduction as:
>
> > However, this combination (BPTT + direct encoding) introduces varying requirements for temporal dependencies across the network spatial dimension—an aspect overlooked by existing research…. (Starting from line 54)
>
> Therefore, the design of our approach and the targeted performance improvements are inherently tied to the BPTT + direct encoding methodology. To the best of our knowledge, long time-step memory tasks do not employ direct encoding methods, that is why we did not choose this dataset. We attempted to implement our method on the S-MNIST dataset reported in the paper you mentioned to evaluate its performance. However, since our approach relies on BPTT, and the S-MNIST data requires T=784 time steps for full representation, the runtime for BPTT becomes unacceptably long. This can be inferred from the BPTT backward propagation equation as follows,
>
> \begin{equation}
>     \nabla_{\mathbf{X}}\mathbf{S} = \sum_{t=0}^T \sum_{i=0}^t \frac{\partial S[t]}{\partial V[i]} \frac{\partial V[i]}{\partial X[i]} = \sum_{t=0}^T \Theta'(V[t]-V_\text{th}) \cdot (1+\sum_{i=0}^t \exp(-\frac{i}{\tau}) \cdot \prod_{j=1}^i \kappa_h (V[t-j]))
> \end{equation}
>
>
> where it involves T consecutive multiplications. Consequently, we were unable to provide corresponding performance results within the limited rebuttal period, and we apologize for this. Nonetheless, your suggestion presents a promising direction for future work, and we express our gratitude for this valuable insight.

---

> ### Comment · Reviewer_B74X · 2024-11-20
>
> The number of time-steps in the sequential classification task is the width of the image, e.g., T=32 for the sequential CIFAR10. I think that T=32 is not too large for BPTT?

---

> > ### Author Response · Authors · 2024-11-23
> > **Response to to Reviewer B74X**
> >
> > Thank you for your response and valuable suggestion. Indeed, as you mentioned, unfolding a 32x32 CIFAR image into a sequence of 32x32x1 is not particularly challenging for BPTT. Since the input data dimension for each time step changes from 32x32 to 32x1, in our implementation of Sequential CIFAR-10, we replaced `nn.Conv2d`, `nn.BatchNorm2d`, and `nn.AvgPool2d` in the program with `nn.Conv1d`, `nn.BatchNorm1d`, and `nn.AvgPool1d`, respectively, while leaving other network structures unchanged.
> >
> > Below, we present the optimal architectures and their performance on Sequential CIFAR-10 using ResNet-18 and VGG-11:
> >
> > | Arc. | timestep | batchsize | epoch | Configuration | Acc. (%) |
> > |--|--|--|--|--|--|
> > | ResNet-18 | 6 | 128 | 200 | 10201110001000111 | 76.73% |
> > | VGG-11 | 6 | 128 | 200 | 20110111 | 75.45% |
> >
> > We attempted to find SNN studies that perform similar row-wise processing on CIFAR-10 for comparison. However, we regret to inform you that we have not yet found examples of SNNs applied to row-wise CIFAR-10 in SOTA methods. Instead, we turned to research on RNNs in this area and found that existing studies differ in their definitions of Sequential CIFAR-10. For instance, Section 5.2 of [1] and Section 4 of [2] explicitly define Sequential CIFAR-10 as pixel-by-pixel unfolding, requiring 32x32=1024 time steps to represent the data.
> >
> > Currently, among the studies we have identified, [3] defines Sequential CIFAR-10 with row-wise unfolding in Section 4.4, aligning with our approach of unfolding the data by rows instead of individual pixels. The accuracy reported by [3] for row-wise processing of CIFAR-10 is 71.03%.
> >
> > > [1] Chang, Bo, et al. "AntisymmetricRNN: A dynamical system view on recurrent neural networks." ICLR, 2019
> >
> > > [2] Kozachkov, Leo, et al. "RNNs of RNNs: Recursive construction of stable assemblies of recurrent neural networks. " NeurIPS, 2022.
> >
> > > [3] Kag, Anil, and Venkatesh Saligrama. "Training recurrent neural networks via forward propagation through time." ICML, 2021.

---

> > > ### Comment · Reviewer_B74X · 2024-11-24
> > >
> > > Hi, thanks for the new results! But the accuracy is not good, which might be caused by the network structure. And your timestep is 6, rather than 32. Is it an error?
> > > You can refer to this paper: `Parallel Spiking Neurons with High Efficiency and Ability to Learn Long-term Dependencies`, which reports the results of PSN/GLIF/PLIF/LIF and provide source codes for sequential CIFAR10/CIFAR100.

---

> > > > ### Author Response · Authors · 2024-11-25
> > > > **Response to to Reviewer B74X**
> > > >
> > > > Thank you for your response and suggestions. First, please allow us to express our apologies for the error in our previous reply, where we mistakenly stated the timestep as 6 instead of 32. After thoroughly reviewing the program we used, we have confirmed that both the search and training phases were conducted with a timestep of 32, not 6. Consequently, the accuracy reported in the previous reply reflects the actual results we obtained.
> > > >
> > > > While the previously reported performance was not particularly outstanding, we would like to first present the accuracy achieved **without** applying our method. This baseline performance, obtained using the ResNet-18 and VGG-11 network frameworks, is provided to demonstrate the extent of performance improvement achieved with our method.
> > > >
> > > > | Arc. | Neuron | Epoch | Acc. (%) |
> > > > |--|--|--|--|
> > > > | ResNet-18 | LIF | 200 | 70.79 |
> > > > | VGG-11 | LIF | 200 | 68.02 |
> > > >
> > > > Compared to the performance reported in our previous reply (ResNet-18: 76.73%, VGG-11: 75.45%), it is evident that our method significantly enhances network performance, achieving improvements of 5.94% on ResNet-18 and 7.43% on VGG-11.
> > > >
> > > > However, the accuracy we reported still lags considerably behind the performance presented in [1], which can be attributed to several factors.
> > > >
> > > > * First, the additional experiments we conducted have not undergone extensive tuning. As a result, the hyperparameter selection and network architecture design for our method may not yet be optimized.
> > > >
> > > > * Second, we did not utilize the network framework employed in [1]. We believe that the inherent properties of ResNet and VGG could limit their performance on sequential datasets. The network framework used in [1] is CIFARNET, a 7-layer network, which aligns with the framework adopted in [2].
> > > >
> > > > * Third, we did not implement the data augmentation techniques reported in [1], which further constrained the network’s performance.
> > > >
> > > > Based on the above considerations, we modified the network framework in our program to align with CIFARNET as used in [1]. Additionally, we adopted the training parameters specified in [2], such as using cross-entropy loss (instead of TET loss), setting the number of epochs = 256, and applying the data augmentation techniques reported in [1]. These included random cutmix [3] with $p = 1$, $a = 1$, trivial augment [4], random erasing [5] with $p = 0.1$, and label smoothing [6] with an amount of $0.1$.
> > > >
> > > > To ensure the accuracy of our reproduction, we evaluated the baseline performance using only LIF neurons for comparison. The experimental results are presented in the table below. Notably, in the Seq-CIFAR-100 dataset, we conducted two sets of experiments of our method. This was due to oscillations in the maximum architecture parameter for the fourth layer neurons during the final stage of the search (last 20 epochs), where the parameter alternated repeatedly between neuron types LLIF (1) and SLIF (2). Consequently, we trained models for both architectures.
> > > >
> > > > | dataset | Neuron / Configuration | Acc. (%) |
> > > > |--|--|--|
> > > > | Sequential CIFAR-10 | LIF | 80.84 |
> > > > | | 1011011 | 87.25 ($\uparrow$ 6.41) |
> > > > | Sequential CIFAR-100 | LIF | 55.32 |
> > > > | | 1002111| 60.54 ($\uparrow$ 5.22) |
> > > > | | 1001111| 60.73 ($\uparrow$ 5.41) |
> > > >
> > > > From the results above, it is evident that our method outperforms the LIF baseline by 6.41% and 5.41% on Sequential CIFAR10 and CIFAR100, respectively, demonstrating a considerable enhancement in network performance on sequential datasets. On the other hand, while our method slightly underperforms the accuracy reported in [1] (CIFAR10, PSN: 88.45%; CIFAR100, PSN: 62.21%), it still holds significant potential for further optimization. For instance, fine-tuning the network structure or optimizing hyperparameters could potentially narrow the performance gap or even surpass the SOTA methods.
> > > >
> > > > Additionally, the experimental results indicate that neurons primarily utilize neurons of LIF (0) and LLIF (1), a phenomenon similar to that observed in neuromorphic datasets (Sec. 5.1). This suggests that SLIF (2) neurons play a limited role in such long-sequence tasks. However, this does not imply that SLIF neurons are without merit. In our response to reviewer m9Jx, we include a comprehensive set of ablation experiments to demonstrate the necessity of SLIF neurons.
> > > >
> > > > > [1] Fang, Wei, et al. "Parallel spiking neurons with high efficiency and ability to learn long-term dependencies." NeurIPS, 2024.
> > > >
> > > > > [2] Fang, Wei, et al. "Incorporating learnable membrane time constant to enhance learning of spiking neural networks." ICCV, 2021.
> > > >
> > > > > [3] Yun, Sangdoo, et al. "Cutmix: Regularization strategy to train strong classifiers with localizable features." ICCV, 2019.
> > > >
> > > > > [4] Müller, Samuel G. et al. "Trivialaugment: Tuning-free yet state-of-the-art data augmentation." ICCV, 2021.
> > > >
> > > > > [5] Zhong, Zhun, et al. "Random erasing data augmentation." AAAI, 2020.
> > > >
> > > > > [6] Szegedy, Christian, et al. "Rethinking the inception architecture for computer vision." CVPR, 2016.

---

> > > > ### Author Response · Authors · 2024-11-29
> > > > **Response to to Reviewer B74X**
> > > >
> > > > We greatly appreciate your insightful feedback once again. With the discussion period ending soon (Monday, December 2nd), we would like to confirm whether our revisions and responses have sufficiently resolved your concerns. If there are any remaining questions, we’re more than happy to provide further clarification or discuss them in detail.

---

### Official Review · Reviewer_XRkd · 2024-11-02

**Soundness:** 2
**Presentation:** 3
**Contribution:** 2
**Rating:** 6
**Confidence:** 4

**Summary:**

This paper proposes a novel method called Spatio-Temporal Dependency-Aware Neuron Optimization (ST-DANO) to adapt neuron behavior to account for different temporal dependencies across network layers. This is achieved by introducing two neuron types, Long-LIF (LLIF) and Short-LIF (SLIF), which a improve the model’s ability to capture long-term and short-term dependencies, respectively, across different network layers. The authors validate the effectiveness of this approach through extensive experiments, demonstrating state-of-the-art performance improvements on both static and neuromorphic datasets​.

**Strengths:**

The introduction of a novel neuron optimization method that adapts temporal dependencies for SNNs.
##
Significant performance improvements over existing models on both static and neuromorphic datasets.

**Weaknesses:**

The distinction between LLIF and SLIF is redundant. LLIF alters the long-term dependency capabilities of neurons by adjusting the threshold; consequently, it should also be able to modify the short-term dependency capabilities through threshold adjustment, transforming LLIF into SLIF. Adjusting the membrane potential constant and the threshold is not contradictory.
##
The method has not been tested on tasks involving long continuous time-series data (like SMNIST), which would further validate the generalizability of the findings​.

**Questions:**

Why not combine LLIF and SLIF into one?  I suggest the authors provide experiments or analyses to demonstrate the necessity of distinguishing these two types of neurons.
##
Could you analyse the main differences between your model and other LIF variants in terms of neuron design?
##
How much ST-DANO requires extra training and inference cost, such as number of parameters, or FLOPs, compared to a baseline LIF network on the same tasks?

---

> ### Author Response · Authors · 2024-11-19
> **Response to Reviewer xRkd (Part 1/3)**
>
> We sincerely appreciate your thorough review of our work and the constructive suggestions you provided to enhance its quality. Please find our responses below.
>
> **Weakness1 & Question 1: The distinction between LLIF and SLIF is redundant. LLIF alters the long-term dependency capabilities of neurons by adjusting the threshold; consequently, it should also be able to modify the short-term dependency capabilities through threshold adjustment, transforming LLIF into SLIF. Adjusting the membrane potential constant and the threshold is not contradictory. Why not combine LLIF and SLIF into one? I suggest the authors provide experiments or analyses to demonstrate the necessity of distinguishing these two types of neurons.**
>
> Yes, strengthening long- and short-term dependencies can indeed be achieved through either adaptive thresholds or adaptive membrane time constants alone. However, in designing these two types of neurons, we also took into account computational burden and the final model’s energy consumption. Firstly, adaptively increasing the membrane potential threshold intuitively suppresses spike firing while achieving our intended goals. Similarly, when we lower the membrane time constant, it increases the leakage effect of the neuron, which also suppresses spike firing. We explicitly highlighted these two reasons in lines 189-192 of the main text, as follows:
>
> > From the perspective of firing rate, we enhance long-term dependencies by adaptively increasing the membrane potential threshold and strengthen short-term dependencies by reducing the membrane time constant. Both strategies work to suppress spike firing, which not only improves network performance but also reduces energy consumption.
>
> We provide additional experiments focusing on cases with only adaptive membrane potential thresholds or only adaptive membrane time constants. The neuron designs for this experiment are as follows:
>
> First, we redesigned two types of neurons, named VthDown and TauUp:
>
> * VthDown: This corresponds to LLIF, but instead of adaptively increasing the membrane potential threshold, we adaptively decrease it, thereby strengthening the neuron's short-term dependencies. VthDown follows the adaptation rule specified by Eq. 7, with the "+" sign replaced by a "-".
>
> * TauUp: This corresponds to SLIF, but instead of adaptively lowering the membrane time constant, we adaptively increase it, thereby enhancing the neuron's long-term dependencies. TauUp follows the adaptation rule specified by Eq. 9, with the "-" sign replaced by a "+".
>
> Next, we designed three additional control experiments on CIFAR-10 with ResNet-18, using a time step setting of 6, as follows:
>
> * A: Only adaptive membrane potential threshold neurons (LLIF and VthDown) participate in the search, with each enhancing long-term and short-term dependencies, respectively.
>
> * B: Only adaptive membrane time constant neurons (TauUp and SLIF) participate in the search, with each enhancing long-term and short-term dependencies, respectively.
>
> * C: Neurons TauUp and VthDown participate in the search, with each enhancing long-term and short-term dependencies, respectively.
>
> The experimental results are as follows, where long-term dependency neurons are marked as 1, short-term dependency neurons as 2, and LIF neurons as 0 in the optimal neuron configurations:
>
> |Case| Neurons         | Configurations          | Acc. (%) | Avg. fire rate (%) |
> |---|-----------------|-------------------------|----------|--------------------|
> |A|  LLIF, VthDown | 02200000000010010     | 94.52    | 12.98              |
> |B|  TauUp, SLIF   | 20000100200020010     | 93.80    | 12.27              |
> |C|  TauUp, VthDown| 22202010000000011      | 95.01    | 13.26              |
> |Text|  LLIF, SLIF | 20000000000001011    | 95.65    | 12.04              |
> |Text|  LIF       | /                     | 93.46    | 12.17              |
>
> From these experiments, we observe that using only adaptive threshold or only adaptive time constant methods can optimize network performance to a certain extent. However, under these methods, the spike firing rate is higher compared to the baseline LIF method. Therefore, when balancing both performance and energy consumption, we believe that our combined LLIF+SLIF approach is optimal.

---

> ### Author Response · Authors · 2024-11-19
> **Response to Reviewer xRkd (Part 2/3)**
>
> **Weakness 2: The method has not been tested on tasks involving long continuous time-series data (like SMNIST), which would further validate the generalizability of the findings.**
>
> Thank you for this suggestion. The design of our method is based on the BPTT and direct encoding approach, since the SMNIST dataset does not utilize direct encoding, it was not chosen in our experiments. We tried to apply our method to the SMNIST dataset; however, because SMNIST requires 784 timesteps for its representation, the BPTT process becomes extremely time-consuming. We apologize for being unable to complete this within the limited rebuttal period. Nonetheless, your suggestion has provided a valuable direction for our future work, and we sincerely thank you for your thoughtful feedback.
>
> **Question 2: Could you analyse the main differences between your model and other LIF variants in terms of neuron design?**
>
> Thank you for your suggestion. The key features of our neuron design can be summarized as follows:
>
> 1. **Selective Neuron Configuration for Optimal Performance**: Our approach does not aim to replace all neurons in the network with a single type. Instead, it combines different neuron types with a neuron search strategy to achieve optimal overall network configuration. This strategy avoids the limitations that may arise from a singular neuron design and allows the network to be more flexible and efficient in handling different temporal dependencies and task requirements.
>
> 2. **No Additional Hyperparameters Introduced**: We have avoided introducing extra hyperparameters, thereby keeping the model complexity at a low level and reducing the difficulty of model training and tuning. This design ensures the simplicity of our model and minimizes potential hyperparameter search costs, making the model more user-friendly and efficient in practical applications.
>
> 3. **Energy Efficiency with Minimal Additional Computation**: While there is a slight increase in computational load due to our neuron design, this is offset by the suppression of spike firing, resulting in lower overall energy consumption compared to conventional LIF neurons. Detailed data supporting this claim is provided in the response to the next question.
>
> 4. **Inspiring Broader Research Directions**: Our combination of neuron design and search strategy offers a valuable perspective that we hope will inspire SNN researchers to not only focus on temporal dependencies but also explore how neuron design and search can achieve better balance in other network properties that vary along spatial dimensions. We believe that such approaches can promote the development of more flexible and adaptable network architectures.
>
> 5. **Applicability Beyond Our Specific Neurons**: We also believe that similar neuron search strategies could be applied to other LIF variants, rather than simply replacing neuron types across the network, potentially yielding superior performance. Our work is available as an open-source project (**codes are provided in supplementary materials and will be made publicly available after publication**), and we sincerely welcome other researchers to build upon and extend our approach, contributing to the advancement of the field.

---

> ### Author Response · Authors · 2024-11-19
> **Response to Reviewer xRkd (Part 3/3)**
>
> **Question 3: How much ST-DANO requires extra training and inference cost, such as number of parameters, or FLOPs, compared to a baseline LIF network on the same tasks?**
>
> Thank you for your valuable suggestion. Our method does not introduce any additional parameters, ensuring that its theoretical FLOPs remain the same as the baseline LIF model. In the Appendix, we provided a comparison of the spike firing rates between our model and the baseline LIF. Here, we utilize theoretical calculations of accumulation (ACs) and multiply-and-accumulate (MACs) operations using the methods described in [1] and [2], whose detailed definitions can be found in the referenced literature. In these calculations, a 32-bit floating-point AC operation consumes 0.9$pJ$ of energy per operation, while a 32-bit floating-point MAC operation consumes 4.6$pJ$ of energy per operation.
>
> The table below shows the theoretical ACs, MACs, and inference energy consumption of our method. While our model's MACs are slightly higher than those of the baseline LIF model due to the use of LLIF or SLIF neurons in some network layers and the additional computations introduced during the adaptive process, our model achieves lower ACs by suppressing spike firing. In terms of energy consumption, our model exhibits only a 0.15$\mu J$ increase over the conventional approach on the Tiny-ImageNet dataset, while showing lower inference energy consumption on other datasets. Furthermore, when our method is deployed on neuromorphic hardware, the extra MACs introduced by LLIF and SLIF neurons can be optimized using lookup tables, which replace costly arithmetic operations with precomputed tables, further reducing power consumption.
>
> In summary, this analysis demonstrates that our method not only enhances network performance but also reduces energy consumption compared to conventional methods.
>
> | Dataset       | Model | Arc.      | Timestep | ACs(M) | MACs(M) | Energy($\mu J$) |
> |---------------|-------|-----------|----------|--------|---------|------------|
> | CIFAR 10      | **Ours**  | ResNet18  | 6        | 83.95  | 2.93    | 89.03      |
> | CIFAR 100     | **Ours** | ResNet18  | 6        | 78.71  | 2.93    | 84.32      |
> | CIFAR10, 100  | LIF   | ResNet18  | 6        | 84.86  | 2.89    | 89.67      |
> | Tiny-ImageNet | **Ours**  | VGG13     | 6        | 130.64 | 279.29  | 1402.31    |
> |               | LIF   | VGG13     | 6        | 135.25 | 278.84  | 1404.39    |
> | DVS-CIFAR-10  | **Ours**  | VGG11     | 10       | 22.65  | 1080.51 | 4990.73    |
> |               | LIF   | VGG11     | 10       | 25.09  | 1080.00 | 4990.58    |
> | DVS-Gesture   | **Ours**  | VGG11     | 20       | 10.99  | 153.06  | 713.97     |
> |               | LIF   | VGG11     | 20       | 14.65  | 152.50  | 714.67     |
>
> > [1] Han, Song, et al. "Learning both weights and connections for efficient neural network." NeurIPS, 2015.
>
> > [2] Huang, Yulong, et al. "Clif: Complementary leaky integrate-and-fire neuron for spiking neural networks." ICML, 2024.

---

> > ### Comment · Reviewer_XRkd · 2024-11-24
> >
> > Thank you for your response. Most of my concerns have been addressed and I would like to increase my score.

---

> > > ### Author Response · Authors · 2024-11-25
> > > **Response to Reviewer xRkd**
> > >
> > > Dear Reviewer,
> > >
> > > We sincerely appreciate your comments, which have been immensely helpful in improving the quality of our work. We greatly look forward to further discussions with you.
> > >
> > > Best Regards
> > >
> > > Paper 8134 Authors

---

### Official Review · Reviewer_EWch · 2024-11-03

**Soundness:** 3
**Presentation:** 3
**Contribution:** 3
**Rating:** 6
**Confidence:** 4

**Summary:**

This paper explores the varying temporal dependency requirements across the spatial dimension of SNNs and identifies the limitations of the LIF neuron in handling these dependencies. Then it designs two types of neurons, LLIF and SLIF, to address this problem. This paper use neuron search to determine where place these neurons. Experiments demonstrate the effectiveness of the proposed method.

**Strengths:**

1. The whole storyline of this work is complete and reasonable.
2. The conducted experiments have shown the effectiveness of LLIF and SLIF.

**Weaknesses:**

1. The performance on CIFAR-100 is relatively low.
2. Table 2 is too wide.
3. See questions.

**Questions:**

1. In Figure 2 a1 and a2, SLIF-4 fails to converge with ResNet-18 on CIFAR-10. Could the authors give explanations on this?
2. The authors say that SLIF captures short-term dependencies. However, in Eq.(9), $\tau[t]$ is decreased when a spike fires. In my opinion, instead of capturing short-term correlations, SLIF overlooks the correlations between different time steps by its mechanism.
3. For LLIF, the authors say ' To achieve stronger long-term dependencies, the contribution from early time steps must have a higher expected proportion in the total gradient compared to standard LIF neurons'. Could the authors give further explanations on this?
4. In table 3, Search Cost is measured in GPU days. Could the authors provide the ratio of search cost to the training time cost?

---

> ### Author Response · Authors · 2024-11-19
> **Response to Reviewer EWch (Part 1/2)**
>
> We sincerely thank the reviewer for their constructive comments. Below, we provide our responses to all the identified weaknesses and issues.
>
> **Weakness 1: The performance on CIFAR-100 is relatively low.**
>
> We appreciate the reviewer's attention to our performance on the CIFAR-100 dataset. We acknowledge that our relative performance on this dataset is slightly lower, achieving 76.25%, while the best performance on CIFAR-100 was achieved by the Joint A-SNN method (77.39%) [1], which utilizes an ANN-to-SNN training approach that is very different from our direct training method based on surrogate gradients. Notably, Joint A-SNN's performance on CIFAR-10 (95.45%) and Tiny-ImageNet (55.39%) is significantly lower than ours, indicating that our direct training strategy demonstrates superior adaptability and generalization on these tasks. Furthermore, within the same training paradigm (direct training with surrogate gradients (SG)), our method exhibits a clear advantage, with the best performance among other SG methods being RecDis-SSF [2], which only achieved 75.48% accuracy. In the future, we will continue to focus on further optimizing our method to achieve satisfactory performance across all datasets.
>
> > [1] Guo, Yufei, et al. "Joint a-snn: Joint training of artificial and spiking neural networks via self-distillation and weight factorization." Pattern Recognition, 2023.
>
> > [2] Wang, Jingtao, et al. "SSF: Accelerating Training of Spiking Neural Networks with Stabilized Spiking Flow." CVPR, 2023.
>
> **Weakness 2: Table 2 is too wide.**
>
> Following this comment, we have reduced the width of Table 2 as much as possible by using abbreviations such as A2S for ANN-to-SNN and SG for surrogate gradient, among others. However, due to the extensive amount of information contained in the table, maintaining its current width might be the most reasonable way to ensure both data completeness and readability. We are open to further suggestions and would be glad to make additional optimizations if recommended.
>
> **Question 1: In Figure 2 a1 and a2, SLIF-4 fails to converge with ResNet-18 on CIFAR-10. Could the authors give explanations on this?**
>
> Thank you very much for your valuable suggestion. We acknowledge that our previous manuscript did not provide sufficient explanation for this phenomenon, which was an oversight on our part. Therefore, in the revised version, we have added a detailed analysis of this phenomenon at lines 325-329, which is highlighted. The specific content is as follows:
>
> > This is because the closer the network is to the output layer, the stronger its temporal dynamics become, and the more abstract its temporal features. As such, near-output neurons need to integrate contextual information through long-term dependencies. If these neurons are forced to emphasize short-term dependencies, as seen with SLIF-4, it can severely damage network performance.
>
> **Question 2: The authors say that SLIF captures short-term dependencies. However, in Eq.(9), \tau[t] is decreased when a spike fires. In my opinion, instead of capturing short-term correlations, SLIF overlooks the correlations between different time steps by its mechanism.**
>
> Thank you for your valuable feedback. We understand your concern regarding the SLIF model's ability to capture short-term dependencies. Indeed, during forward propagation, SLIF accelerates neuronal leakage by reducing the membrane time constant, which might appear to diminish correlations across different timesteps. However, we wish to emphasize that the true purpose of this mechanism is to enable neurons to respond more quickly to new input signals, thereby better capturing dynamic changes in inputs over the short term.
>
> More importantly, in the backpropagation phase, our model does not ignore these temporal correlations. In fact, during the BPTT process, all historical dependencies are computed and updated. This means that while the membrane time constant adaptively decreases during forward propagation, the flow of gradients during backpropagation captures the dependencies between different timesteps. Therefore, rather than completely disregarding temporal correlations, the SLIF model integrates the contributions of different timesteps comprehensively during the backpropagation phase.

---

> ### Author Response · Authors · 2024-11-19
> **Response to Reviewer EWch (Part 2/2)**
>
> **Question 3: For LLIF, the authors say ' To achieve stronger long-term dependencies, the contribution from early time steps must have a higher expected proportion in the total gradient compared to standard LIF neurons'. Could the authors give further explanations on this?**
>
> Thank you for your valuable suggestion; it has greatly helped us improve the completeness of our article. To explain more clearly the logical relationship between "increased gradient contributions from earlier timesteps" and "long-term dependency," we need to revisit the definition of long-term dependency: it refers to the ability of a neural network to effectively capture and utilize the influence of distant past inputs on the current state when processing sequential data. However, deep networks, especially SNNs with recurrent structures, often face the challenge of gradient decaying when handling long sequences. As shown in the BPTT gradient formula (Eq. 4 in the main text), it contains a multiplicative term that is less than 1 across the time dimension. This means that gradients originating from distant timesteps gradually decay during backpropagation, making it difficult for the network to retain and leverage these distant inputs.
>
> Therefore, in our description, we aimed to convey that increasing the gradient contributions from earlier timesteps imposes a form of "memory bias" on the network, ensuring that it does not easily forget earlier signals when processing long sequences. This is precisely the essence of long-term dependency.
> However, we recognize that this explanation was not sufficiently clear in the original text. Thank you again for pointing out this issue. In the revised manuscript, we have supplemented this explanation as follows, located at lines 227-232 and highlighted for clarity.
>
> > We analyze why LLIF neurons enhance long-term dependencies from the perspective of gradient contribution in Appendix C. To achieve stronger long-term dependencies, the contribution from early time steps must have a higher expected proportion in the total gradient compared to standard LIF neurons. This can alleviate the inherent gradient decay phenomenon in BPTT caused by Eq. (4), thereby helping the network more effectively retain and leverage information from distant time steps during the backpropagation process.
>
> **Question 4: In table 3, Search Cost is measured in GPU days. Could the authors provide the ratio of search cost to the training time cost?**
>
> Thank you for your suggestion. The reason we listed the search time in terms of GPU days in Table 3, rather than providing a ratio of time costs, is that absolute search time more intuitively reflects the time cost associated with our search method. This provides prospective researchers with a clearer understanding of the resource requirements under different conditions, allowing them to evaluate whether our approach is suitable for their specific application scenarios or experimental environments.
>
> In addition, we provide the ratio between the search time and the single training time of the final model for context.
>
> |Dataset     | Timestep | Search (S) | Train (T) | ratio=S/T |
> |------------|-------|------------|-----------|-----------|
> | Cifar 10, 100   | 2     | 0.13       | 0.06      | 2.17      |
> |            | 4     | 0.16       | 0.11      | 1.45      |
> |            | 6     | 0.19       | 0.15      | 1.27      |
> |            | 8     | 0.22       | 0.18      | 1.22      |
> | Tiny-ImageNet      | 2     | 0.37       | 0.22      | 1.68      |
> |            | 4     | 0.48       | 0.37      | 1.30      |
> |            | 6     | 0.59       | 0.54      | 1.09      |
> |            | 8     | 0.70       | 0.78      | 0.90      |
> | Dvs-Cifar-10   | 8     | 0.11       | 0.09      | 1.22      |
> |            | 10    | 0.13       | 0.09      | 1.44      |
> | Dvs-Gesture       | 8     | 0.14       | 0.04      | 2.50      |
> |            | 16    | 0.17       | 0.08      | 2.12      |
> |            | 20    | 0.21       | 0.09      | 2.33      |
>
> The data above indicates that NAS methods indeed require more computational resources and time costs compared to conventional training. This is an inherent aspect of NAS approaches, as they involve extensive evaluations of candidate networks and hyperparameter searches.

---

> > ### Comment · Reviewer_EWch · 2024-11-27
> >
> > Thank you for your response. Most of my concerns have been addressed and I would like to keep my score.

---

> > > ### Author Response · Authors · 2024-11-28
> > > **Response to Reviewer EWch**
> > >
> > > Dear Reviewer,
> > >
> > > Thank you very much for your suggestions and recognition. We look forward to further communication with you.
> > >
> > > Paper 8134 Authors

---

### Official Review · Reviewer_m9Jx · 2024-11-03

**Soundness:** 2
**Presentation:** 3
**Contribution:** 2
**Rating:** 6
**Confidence:** 5

**Summary:**

This paper introduces a novel Spatio-Temporal Dependency-Aware Neuron Optimization (ST-DANO) method for spiking neural networks (SNNs). The ST-DANO approach aims to address the limitations of uniform temporal dependency configurations in SNNs by adapting to varying temporal dependencies across the network's layers. Specifically, the authors propose two neuron variants, Long-LIF and Short-LIF, which are designed to capture long-term and short-term dependencies, respectively. These neuron types dynamically adjust membrane potential thresholds and time constants, enhancing the model’s adaptability to temporal variations in data. The authors validate their approach through ablation studies and layer-wise neuron selection strategies. Experimental results on static and neuromorphic datasets demonstrate that ST-DANO achieves notable improvements in accuracy, with a reported performance of 83.90% on the DVS-CIFAR-10 dataset, which is more than a 5% improvement over the baseline.

**Strengths:**

The paper addresses an important issue in SNNs by optimizing neuron behavior based on temporal dependencies, which could enhance the adaptability and performance of SNNs in time-sensitive applications. The introduction of Long-LIF and Short-LIF neurons is a valuable attempt to balance long-term and short-term dependencies in the network. The experimental results, particularly on neuromorphic datasets, demonstrate the potential of the proposed approach, showing notable improvements over baseline models.

**Weaknesses:**

There exists extensive research on neuron design that dynamically adjusts neuron parameters, such as thresholds and membrane potential constants, to optimize SNN performance. However, the authors did not review or compare their approach against this body of work, nor do they demonstrate clear advantages over these established methods. For example, previous studies have explored biologically inspired dynamic thresholds (Ding et al., 2022), synapse-threshold synergistic learning (Sun et al., 2023), and adaptive threshold plasticity (Zhang et al., 2023), which dynamically adjust neuron properties to improve network adaptability and efficiency:
1.	Ding J, Dong B, Heide F, et al. "Biologically inspired dynamic thresholds for spiking neural networks." Advances in Neural Information Processing Systems, 2022, 35: 6090-6103.
2.	Sun H, Cai W, Yang B, et al. "A synapse-threshold synergistic learning approach for spiking neural networks." IEEE Transactions on Cognitive and Developmental Systems, 2023, 16(2): 544-558.
3.	Zhang A, Shi J, Wu J, et al. "Low latency and sparse computing spiking neural networks with self-driven adaptive threshold plasticity." IEEE Transactions on Neural Networks and Learning Systems, 2023.
4.	Hassan A, Meng J, Seo J S. "Spiking Neural Network with Learnable Threshold for Event-based Classification and Object Detection." 2024 International Joint Conference on Neural Networks (IJCNN), IEEE, 2024: 1-8.
5.	Wei W, Zhang M, Qu H, et al. "Temporal-coded spiking neural networks with dynamic firing threshold: Learning with event-driven backpropagation." Proceedings of the IEEE/CVF International Conference on Computer Vision, 2023: 10552-10562.
6.	Hao Z, Shi X, Pan Z, et al. "LM-HT SNN: Enhancing the Performance of SNN to ANN Counterpart through Learnable Multi-hierarchical Threshold Model." arXiv preprint arXiv:2402.00411, 2024.
7.	Wang S, Cheng T H, Lim M H. "LTMD: learning improvement of spiking neural networks with learnable thresholding neurons and moderate dropout." Advances in Neural Information Processing Systems, 2022, 35: 28350-28362.
8.	Yao X, Li F, Mo Z, et al. "Glif: A unified gated leaky integrate-and-fire neuron for spiking neural networks." Advances in Neural Information Processing Systems, 2022, 35: 32160-32171.
Furthermore, significant advancements have been made in the evolution of neuron and circuit design for SNNs, which allow networks to adaptively evolve neuron properties to suit different tasks and data distributions. Notable studies in this area include:
9.	Shen G, Zhao D, Dong Y, et al. "Brain-inspired neural circuit evolution for spiking neural networks." Proceedings of the National Academy of Sciences, 2023, 120(39): e2218173120.
10.	Che K, Zhou Z, Yuan L, et al. "Spatial-Temporal Search for Spiking Neural Networks." arXiv preprint arXiv:2410.18580, 2024.
11.	Zhang A, Han Y, Niu Y, et al. "Self-evolutionary neuron model for fast-response spiking neural networks." IEEE Transactions on Cognitive and Developmental Systems, 2021, 14(4): 1766-1777.
12.	Shen J, Xu Q, Liu J K, et al. "Esl-snns: An evolutionary structure learning strategy for spiking neural networks." Proceedings of the AAAI Conference on Artificial Intelligence, 2023, 37(1): 86-93.
The proposed method in this paper relies on predefined neuron types (Long-LIF and Short-LIF) and does not explore a generalized strategy that could dynamically balance temporal dependencies within the network. This reliance on fixed neuron types limits the flexibility and adaptability of the method, especially compared to recent approaches that evolve neuron designs or dynamically adjust neuron parameters. The study also lacks comparisons with state-of-the-art methods in Tab.2 and provides limited ablation studies on neuron combinations (such as only LIF and SLIF, LIF and LLIF, SLIF and LLIF) and verified on other encoding paradigms. Additionally, the method has not been validated on complex architectures, such as RNNs or Transformers, nor on large-scale datasets like ImageNet, which limits its generalizability and practical applicability.

**Questions:**

1.	Could the authors explore a generalized strategy for temporal adaptation that does not rely on predefined neuron types?
2.	How would the proposed ST-DANO method perform with other encoding paradigms?
3.	Could the authors conduct ablation studies to investigate neuron combinations, such as only Long-LIF or only Short-LIF?
4.	Would this approach work effectively on more complex architectures, such as RNNs or Transformers, and on large-scale datasets like ImageNet?

---

> ### Author Response · Authors · 2024-11-19
> **Response to Reviewer m9Jx (Part 1/4)**
>
> We sincerely appreciate your thorough review of our work. We will now address each of the weaknesses and questions you raised individually.
>
> **Weakness 1: There exists extensive research on neuron design that dynamically adjusts neuron parameters, such as thresholds and membrane potential constants, to optimize SNN performance. However, the authors did not review or compare their approach against this body of work, nor do they demonstrate clear advantages over these established methods.**
>
> Thank you for this valuable comment. Here, we would like to clarify the similarities and differences between our work and existing approaches:
>
> 1. Due to space constraints, we selected a few representative cases from the literature you listed [1-3], which contained relatively comprehensive experiments, and incorporated them into our performance comparison, specifically in Table 2 (lines 383 to 431 in the revised manuscript), where added methods are highlighted in blue. The conclusion drawn is that our method consistently demonstrates advantages across various datasets.
>
> > [1] Hasssan, Ahmed, Jian Meng, and Jae-Sun Seo. "Spiking Neural Network with Learnable Threshold for Event-based Classification and Object Detection."  IJCNN, 2024
>
> > [2] Wei, Wenjie, et al. "Temporal-coded spiking neural networks with dynamic firing threshold: Learning with event-driven backpropagation." ICCV, 2023.
>
> > [3] Shen, Jiangrong, et al. "Esl-snns: An evolutionary structure learning strategy for spiking neural networks."AAAI, 2023.
>
> 2. It is true that considerable research has been conducted on optimizing SNN performance by dynamically adjusting neuron hyperparameters such as thresholds and membrane time constants. However, the focus of the "Related Work" section in our review was not on these conventional dynamic adjustment studies, but rather on examining the temporal dependency issue in SNNs. This focus aligns with the core goal of our work, which is to enhance long-term and short-term temporal dependencies through LLIF and SLIF, rather than simply adjusting conventional parameters. We believe that the review of temporal dependency issues is relatively less explored in existing literature, thereby positioning our work as both unique and of significant value in this domain.
>
> **Weakness 2: The study also lacks comparisons with state-of-the-art methods in Tab.2**
>
> Thank you for your feedback. However, please allow us to clarify that the works listed in Table 2 including and not limited to Online LTL (2022) [1], Joint A-SNN (2023) [2], TET-SSF (2023) [3], OTTT (2022) [4], SLTT (2023) [5], and sparse-KD (2024) [6], etc., represent the latest SNN research, all published between 2022 and 2024.
>
> > [1] Yang, Qu, et al. "Training spiking neural networks with local tandem learning." NeurIPS, 2022.
>
> > [2] Guo, Yufei, et al. "Joint a-snn: Joint training of artificial and spiking neural networks via self-distillation and weight factorization." Pattern Recognition, 2023
>
> > [3] Wang, Jingtao, et al. "SSF: Accelerating Training of Spiking Neural Networks with Stabilized Spiking Flow." CVPR, 2023.
>
> > [4] Xiao, Mingqing, et al. "Online training through time for spiking neural networks."  NeurIPS, 2022.
>
> > [5] Meng, Qingyan, et al. "Towards memory-and time-efficient backpropagation for training spiking neural networks." ICCV, 2023.
>
> > [6] Xu, Qi, et al. "Reversing Structural Pattern Learning with Biologically Inspired Knowledge Distillation for Spiking Neural Networks." ACM Multimedia, 2024.

---

> ### Author Response · Authors · 2024-11-19
> **Response to Reviewer m9Jx (Part 2/4)**
>
> **Question 1: Could the authors explore a generalized strategy for temporal adaptation that does not rely on predefined neuron types?**
>
> We sincerely thank the reviewer for the suggestion regarding exploring a general temporal adaptation strategy independent of predefined neuron types. We deeply appreciate this recommendation and recognize the significant research value and challenges associated with such a strategy.
>
> Firstly, we chose to design specific predefined neuron types such as LLIF and SLIF to purposefully enhance the temporal dependency performance of SNNs. Subsequently, we developed a NAS-based framework to enable diverse dependency configurations within the network. This approach offers a direct method for optimization across different temporal scales, and it remains highly intuitive. In the worst-case scenario, our method can still search for architectures composed entirely of LIF neurons, ensuring that our approach is at least as good as the baseline. Although this method may, to some extent, rely on predefined neuron types, its performance across multiple datasets demonstrates that its generalizability is not limited by these predefined neurons. It has exhibited significant advantages in handling various types of tasks, fully showcasing its adaptability and practical value.
>
> Regarding the reviewer's suggestion to explore a general temporal adaptation strategy, we believe this is a highly valuable research direction. In future work, we plan to investigate how to construct more flexible, data-driven temporal adaptation strategies to achieve time adjustment methods that do not rely on specific neuron types.
>
> **Question 2: How would the proposed ST-DANO method perform with other encoding paradigms?**
>
> Thank you for the suggestion. Our method is specifically designed to address the issue of temporal dependency variations across the spatial dimension brought about by the mainstream training combination of BPTT and direct encoding. This particular context has determined the focus of our current work and guided our methodological design. We have elaborated on the motivation for this work in detail in the Introduction section as follows:
>
> > However, this combination (BPTT + direct encoding) introduces varying requirements for temporal dependencies across the network spatial dimension—an aspect overlooked by existing research…. (Starting from line 54)
>
> Regarding other encoding methods, such as rate coding and temporal coding, the temporal dependency variation issue we described does not exist. Therefore, we must acknowledge that our method may not be perfectly compatible with other encoding methods. Nonetheless, we believe that applying the design concepts of our ST-DANO approach to other encoding methods is a meaningful research direction, as our method intuitively enhances network performance.

---

> ### Author Response · Authors · 2024-11-19
> **Response to Reviewer m9Jx (Part 3/4)**
>
> **Question 3: Could the authors conduct ablation studies to investigate neuron combinations, such as only Long-LIF or only Short-LIF?**
>
> Thank you for your thorough review. In fact, we have already conducted ablation experiments in the original manuscript, but these experiments were placed before the main experiment section, serving as the motivation for our proposed neuron search framework. Specifically, they are located at lines 260-346 (in the revised manuscript) under the section titled "**4.2 Ablation study: Why a Neuron Search Strategy is Essential?**".
>
> In our ablation experiment design, we selected representative datasets and network architectures from both static and neuromorphic data categories—Cifar10 with ResNet-18 and DVS-CIFAR-10 with VGG11. By selectively replacing LIF neurons at different layers with LLIF and SLIF, we tested their performance, including scenarios where all neurons were replaced with either LLIF or SLIF. Detailed experimental results and specifics can be found in Table 1, at lines 287-304.
>
> Additionally, to more comprehensively address the reviewer's suggestions, we have conducted further ablation experiments on additional datasets that were not covered in the main text (CIFAR-100, DVS-CIFAR-10, and DVS-Gesture). In these additional ablation experiments, we used the same hyperparameters as specified in Table 5 of Appendix F. Here, LLIF, SLIF, and LIF indicate configurations where all neurons within the architecture are replaced with LLIF, SLIF, and LIF neurons respectively, while "Text" represents the performance under the optimal neuron configuration presented in our main text.
>
> |     | Arc.     | Timestep | LLIF  | SLIF  | LIF   | Text  |
> |--------------|----------|-------|-------|-------|-------|-------|
> | Cifar 100    | Resnet18 | 6     | 73.20 | 63.97 | 73.03 | 75.94 |
> | Tiny-ImageNet      | VGG13    | 6     | 57.33 | 48.01 | 55.17 | 60.14 |
> | Dvs-Gesture   | VGG11    | 20    | 98.61 | 81.60 | 95.83 | 98.61 |
>
> It can be observed that, generally, the LLIF method enhances performance, while SLIF tends to have a negative impact on performance. The reason why LLIF and "Text" yield the same result for DVS-Gesture is that the optimal configuration reported in the main text (10111111, where 1 represents LLIF and 0 represents LIF) is essentially identical to using only LLIF neurons, hence their performance is nearly the same. Additionally, due to the limited time available for the rebuttal period, we were unable to conduct a comprehensive ablation study with varying neuron configurations at different layers, as done in Section 4.2. We sincerely apologize for this limitation.
>
> Although the above experiments may suggest that SLIF appears to offer limited benefit, this does not mean that SLIF neurons are entirely without value. In fact, these two types of neurons are complementary. To illustrate the necessity of SLIF, we modified the optimal neuron configuration reported in the main text for CIFAR-10 at timestep 6 (Text: 20000000000001011) into two alternative configurations: A (00000000000001011), where the shallowest layer SLIF neuron is replaced with LIF, and B (10000000000001011), where it is replaced with LLIF. We then compared their performance differences as follows:
>
> |Dataset|Case|Configurations|Acc. (%)|
> |----------|------|--------------------|-------|
> | CIFAR-10 | Text | **2**0000000000001011 | 95.65 |
> |          | A    | **0**0000000000001011 | 94.45 |
> |          | B    | **1**0000000000001011 | 94.24 |
>
> On CIFAR-100, we modified the architecture at time step 6 (Text: 20002000000010111) into two alternative configurations: A (00000000000010111), where the relevant SLIF neurons were replaced with LIF, and B (10001000000010111), where they were replaced with LLIF. The experimental results are as follows:
>
> |Dataset|Case|Configurations|Acc. (%)|
> |----------|------|--------------------|-------|
> | CIFAR-100 | Text | **2**000**2**000000010111 | 75.94 |
> |          | A    | **0**000**0**000000010111 | 74.63 |
> |          | B    | **1**000**1**000000010111 | 75.00 |
>
> It is evident that when SLIF is placed in the shallow layers, it makes a significant contribution to the overall network architecture's performance. Conversely, on CIFAR-10, placing LLIF in the shallow layers (Configuration B) resulted in worse performance than using standard LIF neurons (Configuration A). These experiments demonstrate that SLIF neurons, like LLIF neurons, are essential for balancing temporal dependencies and enhancing network performance. Additionally, they illustrate the rationality and necessity of each module within our method.

---

> ### Author Response · Authors · 2024-11-19
> **Response to Reviewer m9Jx (Part 4/4)**
>
> **Question 4: Would this approach work effectively on more complex architectures, such as RNNs or Transformers, and on large-scale datasets like ImageNet?**
>
> Thank you for raising these questions. Our method is based on the design of LIF neurons in SNNs, including LLIF and SLIF neurons, and aims to optimize specific temporal dependency issues in SNNs. Since our work focuses on the unique characteristics of SNNs and given that RNN and Transformer architectures differ significantly from SNNs (While SNNs rely on event-driven, sparse computations with discrete spikes, RNNs and Transformers utilize continuous signal processing and dense, sequential or attention-based operations), the effectiveness of our approach on RNNs or Transformers remains to be explored. Extending our method to more complex network architectures would require extensive research, redesign, and adaptation, all of which are challenging and time-consuming processes.
>
> Regarding large-scale datasets such as ImageNet, training on them poses significant time and resource challenges, making it difficult to accomplish within the limited rebuttal period. Nonetheless, we believe that with proper expansion and optimization, our approach has the potential to perform well on larger datasets, and this represents a key direction for our future work.

---

> > ### Comment · Reviewer_m9Jx · 2024-11-22
> >
> > Thank you for your detailed response and the additional experimental results. However, I would like to emphasize that it is essential to see the performance of your proposed method on spiking Transformer and spiking RNN architectures, as these more complex and advanced frameworks would demonstrate the broader applicability of your approach beyond traditional CNNs. Additionally, while I understand your NAS search space includes LIF, SLIF, and LLIF, I believe it is crucial to provide comparative results for reduced search spaces, such as (LIF, SLIF) and (LIF, LLIF), rather than solely relying on manual replacements of neuron types. Such comparative studies would better justify the necessity of all three predefined neuron types in the search space. Furthermore, defining SLIF and LLIF as explicit operators might restrict the adaptability of the approach. A more flexible and generalized framework could be achieved by allowing LIF neurons to adaptively exhibit the characteristics of SLIF or LLIF through parameter optimization, rather than explicitly predefining these types. This would avoid potential limitations and demonstrate a more robust and versatile methodology.

---

> > > ### Author Response · Authors · 2024-11-23
> > > **Response to to Reviewer m9Jx (Part 1/2)**
> > >
> > > Thank you for your feedback. Below, we address the three points you raised.
> > >
> > > **1. Extension in Spike RNN and Spikformer**
> > >
> > > We fully understand the importance of validating the proposed method within more complex and advanced frameworks, such as Spikformer and Spike RNN, to demonstrate its broad applicability.
> > >
> > > In this work, we chose to base our method on classical network frameworks (e.g., ResNet and VGG) due to their well-established toolchains, robust hardware support, and extensive real-world deployment scenarios. This approach allowed us to focus on designing the method, performing thorough theoretical analysis, and conducting comprehensive experiments across multiple datasets of various types to demonstrate the method's superiority. Additionally, we have made the source code publicly available to ensure reproducibility. Thus, we believe that the research presented in this work is complete and systematic.
> > >
> > > Due to the fundamental structural differences in frameworks such as Spikformer (e.g., the attention mechanism), our method cannot be directly and straightforwardly applied to these architectures. Extending our method to such frameworks would require additional complex custom designs, extensive debugging and optimization, as well as a comprehensive performance validation process. However, given the time constraints of the rebuttal phase, it is not feasible to complete these intricate tasks within a short period.
> > >
> > > Nonetheless, we greatly appreciate your suggestion. This direction of extension is indeed highly significant and will be one of the key focuses of our future research. Once again, thank you for your insightful feedback on our work!
> > >
> > > **2. Additional Ablation Study**
> > >
> > > Thank you very much for your valuable suggestion. In our previous response, we used a manual replacement approach to illustrate the necessity of the neurons, which we acknowledge was somewhat imprecise, and we sincerely apologize for this. As per your recommendation, we conducted additional ablation experiments as detailed below.
> > >
> > > In this ablation study, we validated the method using three static datasets: CIFAR-10 with ResNet-18, CIFAR-100 with ResNet-18, and Tiny-ImageNet with VGG-13.
> > >
> > > For each dataset, we established the following experimental groups:
> > >
> > > 1. LIF-only (no search required)
> > > 2. LLIF-only (no search required)
> > > 3. SLIF-only (no search required)
> > > 4. LIF+LLIF (search required)
> > > 5. LIF+SLIF (search required)
> > > 6. LIF+LLIF+SLIF (search required)
> > >
> > > For these groups, we used the same hyperparameters as those described in Appendix F. The results are presented below, where "0" denotes LIF neurons, "1" denotes LLIF neurons, and "2" denotes SLIF neurons:
> > >
> > > |dataset|Arc.|Timestep|Neuron|Configuration|Acc. (%)| Tips|
> > > |--|--|--|--|--|--|--|
> > > |CIFAR-10|ResNet-18|6|LIF| \ |93.46|Reported in Sec. 4.2|
> > > | | | |LLIF| \ |94.78|Reported in Sec. 4.2|
> > > | | | |SLIF| \ |83.83|Reported in Sec. 4.2|
> > > | | | |LIF+LLIF| 00110000000000011 |95.20| |
> > > | | | |LIF+SLIF| 20202002000000011 |94.01| |
> > > | | | |LIF+LLIF+SLIF| 20000000000001011 | **95.65** | Reported in Sec. 5.1 |
> > > | | | | | | | |
> > > |CIFAR-100|ResNet-18|6|LIF| \ |73.03|Reported in previous response|
> > > | | | |LLIF| \ |73.20|Reported in previous response|
> > > | | | |SLIF| \ |63.97|Reported in previous response|
> > > | | | |LIF+LLIF| 00000000001011111 |74.89| |
> > > | | | |LIF+SLIF| 02002000200000000 |73.09| |
> > > | | | |LIF+LLIF+SLIF| 20002000000010111 | **75.94** | Reported in Sec. 5.1 |
> > > | | | | | | | |
> > > |Tiny-ImageNet|VGG-13|8|LIF| \ |55.94| |
> > > | | | |LLIF| \ |57.81| |
> > > | | | |SLIF| \ |48.69| |
> > > | | | |LIF+LLIF| 0000001101 |59.23| |
> > > | | | |LIF+SLIF|  0002200000 |56.88| |
> > > | | | |LIF+LLIF+SLIF| 0000020101 | **60.24** | Reported in Sec. 5.1 |
> > >
> > > From the experimental results above, it is evident that the performance achieved by searching for the network framework with a combination of LIF+XLIF neurons surpasses that of manually setting all neurons in the network to a single neuron type. Moreover, the combinations of LIF+LLIF and LIF+SLIF demonstrate lower performance across the three datasets compared to the combination of LIF+LLIF+SLIF. This outcome underscores the rationality of our neuron configuration and the necessity of the NAS framework.

---

> > > ### Author Response · Authors · 2024-11-23
> > > **Response to to Reviewer m9Jx (Part 2/2)**
> > >
> > > **3. Concern about Explicit Operators**
> > >
> > > **At the single-neuron level**, our method explicitly defines the characteristics of SLIF and LLIF neurons through rule-based designs. This approach is intended to clearly differentiate the properties of these two neuron types, ensuring their flexible adaptation and independent optimization for various tasks. This explicit design has demonstrated strong controllability and interpretability in our experiments, especially when evaluating the impact of SLIF and LLIF on task-specific performance. To support this design, we provide a comprehensive analysis of the properties and mathematical derivations of SLIF and LLIF neurons in Appendices C and D.
> > >
> > > While the characteristics of SLIF and LLIF are explicitly defined at the single-neuron level, our method employs NAS **at the network framework level** to dynamically select the neuron types for each layer. This data-driven, implicit hierarchical configuration enables the network to automatically optimize the distribution and combination of SLIF and LLIF neurons according to task requirements, endowing the overall framework with greater adaptability and generality. Specifically, the implicit search mechanism in NAS optimizes the neuron types based on performance objectives, avoiding some flexibility limitations imposed by explicit definitions and ensuring the overall flexibility and task adaptability of the network.

---

> > > > ### Comment · Reviewer_m9Jx · 2024-11-24
> > > >
> > > > Thank you for the additional experiments, which effectively addressed my concerns. I will raise my score accordingly.

---

> > > > > ### Author Response · Authors · 2024-11-25
> > > > > **Response to to Reviewer m9Jx**
> > > > >
> > > > > Dear Reviewer,
> > > > >
> > > > > Thank you for your feedback and recognition of our work. We are more than happy to hear any other questions or suggestions you may have and look forward to further discussions.
> > > > >
> > > > >
> > > > > Best Regards
> > > > >
> > > > > Paper 8134 Authors

---

### Author Response · Authors · 2024-12-03
**General Comments**

First of all, we sincerely thank all the reviewers for their valuable feedback and comments. Below, we summarize the major responses and manuscript revisions made during the rebuttal phase.

&nbsp;

### 1. Additional Ablation Studies

Reviewer m9Jx suggested reducing the search space to validate the performance of different neuron combinations. We conducted additional experiments as suggested, an **the results demonstrate that our proposed neuron combination, LIF+LLIF+SLIF, consistently achieves superior performance across multiple datasets, significantly outperforming other combinations.**

&nbsp;

### 2. Necessity of Distinguishing LLIF and SLIF Neurons

Reviewer xRkd raised the question of whether LLIF and SLIF neurons could be merged into a single type. To address this, we extended the fundamental equations of LLIF and SLIF to create two variant neurons, VthDown and TauUp, and evaluated different neuron combinations within the search space. **The experiments reveal that the combination of LLIF and SLIF neurons not only delivers the best performance but also significantly reduces the network's overall spiking rate**, further supporting the necessity of distinguishing between these two neuron types.

&nbsp;

### 3. Energy Consumption During Inference

Following reviewer xRkd's suggestion, we estimated the energy consumption of our model during inference by calculating the spiking rate. **The results show that, apart from a negligible increase of 0.15$\mu J$ in the Tiny-ImageNet task, the energy consumption of our model is significantly lower than that of the LIF baseline across all other tasks.**

&nbsp;

### 4. Feasibility of Deployment on Neuromorphic Hardware

Reviewer B74X expressed concerns about the deployment challenges of exponential operations on neuromorphic hardware. We clarified that the exponential operations in our method are defined over a small domain, allowing efficient implementation using lookup tables on existing neuromorphic hardware such as [Intel Loihi 2](https://download.intel.com/newsroom/2021/new-technologies/neuromorphic-computing-loihi-2-brief.pdf). Furthermore, we provided evidence that **the precision loss introduced by the lookup table has no negative impact on the network's inference performance.**

&nbsp;

### 5. Gradient Analysis of LLIF Neurons

Reviewer B74X expressed concerns regarding the gradient analysis of LLIF neurons in the original manuscript and encouraged us to differentiate Eq. (4) with respect to V_th . Following this suggestion, **we added an additional section (Appendix C) in the revised manuscript, providing a detailed and rigorous derivation and analysis of the gradients.**

&nbsp;

### 6. Performance on Long-Sequence Tasks
Reviewer B74X also inquired about the performance of our method on long-sequence tasks. To address this, we adjusted the network architecture and conducted experiments on Sequential-CIFAR-10 and Sequential-CIFAR-100. **The results show that our method achieves significant improvements of 6.41% and 5.41% over the LIF baseline on these datasets, respectively.**

---

### Meta-Review · Area_Chair_95ZC · 2024-12-21

**Metareview:**

Thanks for the efforts of reviewers and authors. After the discussion, there are still some pending concerns including:
1.	No validation on complex architectures, such as RNNs or Transformers, and on large-scale datasets like ImageNet (Reviewer m9Jx)
2.	Low accuracy on CIFAR100 (Reviewer EWch)
3.	Fail to converge with ResNet-18 on CIFAR-10 (Reviewer EWch)
4.	No validation on long continuous time-series data such as SMNIST and pixel MNIST (Reviewer xRkd, Reviewer B74X)
5.	Costly exponential operations (Reviewer B74X)
As a conclusion, the proposed methods have not shown overwhelming performance advantages as the authors claim in temporal dependency learning ability than previous spiking neurons. The involvement of exponential operations raises the concerns about the implementation of resource-restricted neuromorphic chips, which have not been fully solved by the authors’ response. So, my decision is to reject it.

**Additional Comments On Reviewer Discussion:**

The authors conducted future ablation experiments on more datasets and showed the different preferences of proposed neuron models. These results indicated the necessity of using the NAS method to choose neuron models. Reviewer m9Jx was satisfied with the new results and raised the score. However, the results on large SNN structures and datasets were not reported.

The authors revised the paper by fixing some format errors, explained the concern about the SLIF neuron, and reported more details about the cost of the NAS method. Reviewer EWch was satisfied and raised the score.

The authors provided additional experiments focusing on cases with only adaptive membrane potential thresholds or only adaptive membrane time constants to answer the question of fusing two neuron models from Reviewer xRkd. These results showed the current method (using two kinds of neuron models) was better. They also compared extra training and inference costs with other methods. However, the cost of exponential operation was not considered, which was emphasized by Reviewer B74X.

Both Reviewer xRkd and Reviewer B74X suggested the authors report accuracy on sequential/pixel classification tasks with long time-steps. However, the authors only reported the results on sequential CIFAR10, which were lower than previous SOTA methods. The authors claimed that it is challenging for their methods to handle tasks with large time-steps.

As suggested by Reviewer B74X, the author replaced the explanation of the influence of threshold with quantitative analysis and added a description of the design of the L-LIF model. Reviewer B74X increased the score with this positive amendment.

In general, the authors solved most of the concerns of reviewers during the rebuttal period. However, the suggestions of validating the proposed methods on more difficult tasks, such as training large-scale SNNs on ImageNet and pixel classification with many time-steps, were pending, which are critical to improve the quality of this paper.

---

### Decision · Program_Chairs · 2025-01-22

Reject